# Neuronal death in pneumococcal meningitis is triggered by pneumolysin and RrgA interactions with β-actin

**Mahebali Tabusi**[1,2], **Sigrun Thorsdottir**[1,2], **Maria Lysandrou**[1,2], **Ana Rita Narciso**[1,2], **Melania Minoia**[3], **Chinmaya Venugopal Srambickal**[4], **Jerker Widengren**[4], **Birgitta Henriques-Normark**[1,2], **Federico Iovino**[1,2¤]*

**1** Department of Microbiology, Tumor and Cell Biology, Karolinska Institutet, BioClinicum J7:20, Stockholm, Sweden, **2** Department of Clinical Microbiology, Karolinska University Hospital, Stockholm, Sweden, **3** Department of Molecular Biosciences, The Wenner-Gren Institutet, Stockholm University, Stockholm, Sweden, **4** Department of Applied Physics, KTH Royal Institute of Technology, Stockholm, Sweden

¤ Current address: Department of Neuroscience, Karolinska Institutet, Biomedicum D07, Stockholm, Sweden.
* federico.iovino@ki.se

**Data Availability Statement:** All relevant data are within the manuscript and its Supporting Information files.

## Abstract

Neuronal damage is a major consequence of bacterial meningitis, but little is known about mechanisms of bacterial interaction with neurons leading to neuronal cell death. *Streptococcus pneumoniae* (pneumococcus) is a leading cause of bacterial meningitis and many survivors develop neurological sequelae after the acute infection has resolved, possibly due to neuronal damage. Here, we studied mechanisms for pneumococcal interactions with neurons. Using human primary neurons, pull-down experiments and mass spectrometry, we show that pneumococci interact with the cytoskeleton protein β-actin through the pilus-1 adhesin RrgA and the cytotoxin pneumolysin (Ply), thereby promoting adhesion and invasion of neurons, and neuronal death. Using our bacteremia-derived meningitis mouse model, we observe that RrgA- and Ply-expressing pneumococci co-localize with neuronal β-actin. Using purified proteins, we show that Ply, through its cholesterol-binding domain 4, interacts with the neuronal plasma membrane, thereby increasing the exposure on the outer surface of β-actin filaments, leading to more β-actin binding sites available for RrgA binding, and thus enhanced pneumococcal interactions with neurons. Pneumococcal infection promotes neuronal death possibly due to increased intracellular Ca²⁺ levels depending on presence of Ply, as well as on actin cytoskeleton disassembly. STED super-resolution microscopy showed disruption of β-actin filaments in neurons infected with pneumococci expressing RrgA and Ply. Finally, neuronal death caused by pneumococcal infection could be inhibited using antibodies against β-actin. The generated data potentially helps explaining mechanisms for why pneumococci frequently cause neurological sequelae.

**Funding:** We thank all the major funding that has supported this study: Petrus and Augusta Hedlund Foundation, Jeansson Foundation, Åke Wiberg Foundation, SFO StratNeuro Start-up grant, Clas Groschinsky Foundation, Karolinska Institutet Faculty Board, and the Karolinska Institutet Research Foundation Grants to FI, as well as the Swedish Research Council, the Knut and Alice Wallenberg Foundation, Stockholm County Council, and the Swedish foundation for Strategic research (SSF) to BHN. The funders had no role in study design, data collection and analysis, decision to publish, or preparation of the manuscript.

**Competing interests:** The authors have declared that no competing interests exist.

## Author summary

Neuronal damage is a major consequence of meningitis. *Streptococcus pneumoniae* (pneumococcus) is the leading etiological cause of bacterial meningitis, yet how pneumococci interact with neurons and cause neuronal death is poorly understood. Using human neurons *in vitro* and our established bacteremia-derived meningitis mouse model *in vivo*, we found that pneumococci use the pilus-1 adhesin RrgA and the cytotoxin pneumolysin (Ply) to interact with neuronal β-actin expressed on the plasma membrane. Also, we demonstrate that Ply interaction with the neuronal plasma membrane increase the exposure of β-actin on the neuronal plasma membrane, allowing more pneumococci to adhere to neurons through RrgA-β-actin interaction. Moreover, neurons infected with RrgA- and Ply-expressing pneumococci showed increased intracellular $Ca^{2+}$ levels and disruption of β-actin filaments, possibly leading to neuronal death. Importantly, by blocking pneumococcal-β-actin interaction using antibodies, we could reduce neuronal cell death after pneumococcal infection.

## Introduction

The Gram-positive bacterium *Streptococcus pneumoniae* (the pneumococcus) is the main cause of bacterial meningitis worldwide [1, 2]. Despite access to antibiotics and intensive care, the mortality rate in pneumococcal meningitis remains high, ranging from 10–40% depending on geographical region [3, 4]. Moreover, around 50% of the survivors suffer from permanent neurological damages, sequelae, after the infection has resolved [5–8]. Little is known on mechanisms for how pneumococci interact with and damage neurons.

Pneumococcal infections usually start with pneumococcal colonization of the upper respiratory tract. From this location bacteria may reach the blood stream and then interact and bypass the blood-brain barrier (BBB) endothelium to cause meningitis [9, 10]. It has been shown that the RrgA protein of the pneumococcal pilus-1 interacts with the endothelial receptors pIgR and PECAM-1, thereby promoting pneumococcal spread through the BBB and infection of the brain [1]. Pilus-1, which is present in ∼30% of pneumococcal isolates, has previously been shown to promote colonization, virulence, and pro-inflammatory responses in mouse models [9, 11], and it is composed of three structural proteins, RrgA, RrgB and RrgC, where the tip pilin protein RrgA has been found to be the major adhesin to epithelial cells [12], and RrgB to be the major stalk protein [11]. The cytotoxin of pneumococci, pneumolysin (Ply), has also been shown to affect the brain. Indeed, purified Ply was observed to induce microglial and neuronal apoptosis in the brain [13, 14]. Ply is a cholesterol-dependent and pore-forming toxin that has been found to induce pro-inflammatory responses, but recently we showed that in certain cell types such as alveolar macrophages and dendritic cells, Ply may also induce anti-inflammatory responses and effect T-cell responses [15].

Pneumococci interact with host cells by adhering to their surfaces and by becoming internalized into certain cells such as immune cells. Like other bacteria they may exploit transport systems of host cells as mechanisms for adhesion and uptake. For example, both pneumococcal RrgA and the surface protein PspC have been shown to bind to pIgR (polymeric immunoglobulin receptor), whose physiological function is to transport immunoglobulins across human-cell barriers, on human nasopharyngeal epithelial and brain endothelial cells, and the physiological function of pIgR is to transport immunoglobulins across human-cell barriers [16–18].

In eukaryotic cells, the actin skeleton is essential for cellular development, metabolism, and immunity [19], and β-actin is a predominant isoform among the actin cytoskeleton proteins

[20, 21], essential for cell growth, migration, and the G-actin pool [22, 23]. In neurons, one of the main functions of actin filaments is the transport of microtubules along axons which is fundamental for neuronal development [24]. Some bacterial pathogens, such as *Shigella* and enteropathogenic *Escherichia coli*, have been shown to exploit cytoskeletal proteins in eukaryotic cells to activate host-mediated uptake [25, 26]. Furthermore, the cell wall of Gram-positive bacteria was also previously reported to be internalized by epithelial and endothelial cells through an actin-dependent pathway [27]. However, little is known about pneumococcal interactions with the actin cyto skeleton. It has been demonstrated though that Ply has direct transmembrane interactions with actin within the lipid bilayer of the plasma membrane of astrocytes [28].

Here, we investigated pneumococcal interactions with neurons. We found that the pilus-1 adhesin RrgA can bind to neurons. We show that presence of RrgA and the cytotoxin Ply promote pneumococcal entry into neurons. Also, we found that purified proteins or presence of RrgA and Ply in pneumococcal strains cause neuronal death, and that both proteins interact with β-actin of the neuronal cytoskeleton and induce $Ca^{2+}$ release, thereby potentially damaging the cytoskeleton and the cells, as we find that disruption of β-actin filaments in neurons infected with pneumococci expressing RrgA and Ply was observed using STED super-resolution microscopy. Furthermore, the neurotoxic effect could be blocked using anti-β-actin antibodies.

## Results

### Neuronal cell death caused by a pneumococcal infection is pilus-1 and pneumolysin dependent

We first investigated the importance of pilus-1 and pneumolysin (Ply) in neuronal death. As *in vitro* model for neurons we used SH-SY5Y human neuroblastoma cells, and neurons obtained by differentiating SH-SY5Y cells using retinoic acid as previously described [29]. Validation of the neuronal differentiation was assessed by detection of neuronal markers MAP2 and NSE [30, 31] by western blot and immunofluorescence staining (S1A and S1B Fig) and by morphological microscopy analysis (S1C Fig). Neurons were challenged with the pneumococcal strain TIGR4 and then stained with a live/dead dye and visualized using high-resolution live-cell imaging microscopy. We observed that the neurons were rapidly killed by the TIGR4 strain. Thus, 2 hours post-infection all imaged neurons showed clear signs of cell death (Fig 1A). To study the roles of pilus-1 and Ply, we used mutant strains lacking the pilus-1 islet, TIGR4Δ*rrgA-srtD*, [9] or Ply, TIGR4Δ*ply* [15]. When infected with either of the two mutants, neurons showed a significant decrease in cell death compared with wild-type (wt) TIGR4 infection (S2A and S2B Fig). Furthermore, neuronal cell death caused by TIGR4Δ*ply* was approximately 50% lower than what was observed for TIGR4Δ*rrgA-srtD* (S2C Fig). These data suggest that the neuronal damage caused by the toxin Ply, which could be released after bacterial lysis or while being attached on the pneumococcal surface [14, 15], is more severe than the damage caused by pilus-1, which mediate physical interaction with the host cell. After 2 hours infection with the double mutant TIGR4Δ*rrgA-srtD*Δ*ply*, the neuronal death rate was less than 5% of wt TIGR4, and considerably lower than infection with either of the single mutants TIGR4Δ*rrgA* or TIGR4Δ*ply* (S2A–S2C Fig). A similar trend was also observed when an LDH assay was performed to assess the viability of neurons (S3 Fig). The reduced neuronal cell death upon infection with the pilus-1 deletion mutant TIGR4Δ*rrgA-srtD* was not due to an impaired expression of Ply since a western blot analysis showed similar expression of Ply in TIGR4 as in TIGR4Δ*rrgA-srtD* (S4 Fig). Thus, we conclude that both pilus-1 and Ply contribute to neuronal cell damage.

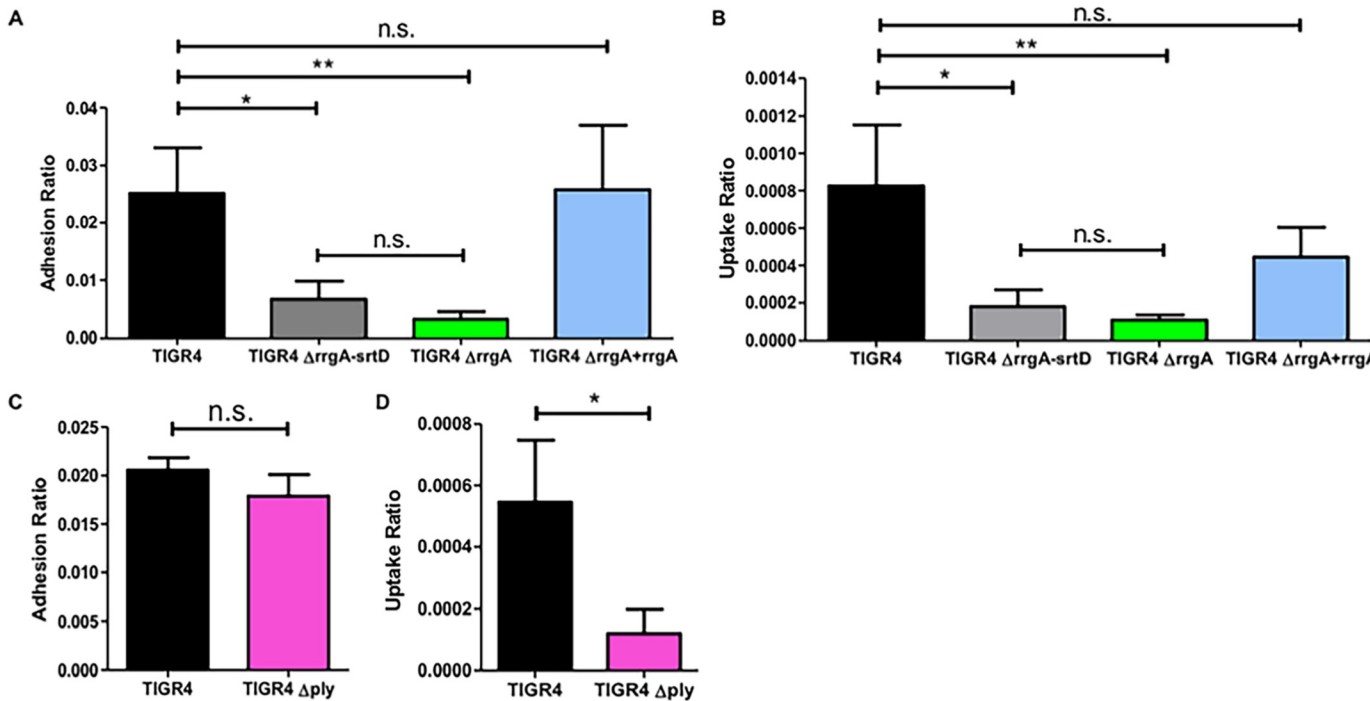

**Fig 1. RrgA promotes pneumococcal binding to neurons, and both RrgA and Ply increases pneumococcal invasion into neuronal cells.** Pneumococcal interaction with neurons was investigated *in vitro* using differentiated neurons and infection with (A and B) wt TIGR4 and its isogenic mutants, TIGR4Δ*rrgA-srtD*, TIGR4Δ*rrgA*, and TIGR4Δ*rrgA+rrgA*, and in (C and D) wt TIGR4 and TIGR4Δ*ply*. (A) CFU-based adhesion to neurons. (B) Invasion into neurons. (C) Adhesion ratio was calculated as [CFU of adhered bacteria] / [CFU of (non−adhered bacteria + adhered bacteria)]. (D) Invasion ratio was calculated as [CFU of invaded bacteria] / [CFU of adhered bacteria)]. For all graphs (A-D) the columns represent average values, and error bars represent standard deviations. Each graph shows an overview at least three (n≥3) biological replicates. ** = p<0.001, * = p<0.05, n.s. = non-significant.

## The pilus-1 adhesin RrgA mediates pneumococcal binding to and invasion of neurons

Next, we studied whether pilus-1 influences pneumococcal adhesion to and internalization by neurons. First, we infected SH-SY5Y cells or neurons with wt TIGR4 and found that pneumococci can adhere to both differentiated neurons and SH-SY5Y before differentiation (Figs 1A and S5A). Then we used our pilus-1 deletion mutant, TIGR4Δ*rrgA-srtD*, and observed that it adhered significantly less to neurons and SH-SY5Y cells as compared to wt TIGR4 (Figs 1A and S5A), suggesting that pilus-1 promotes pneumococcal interaction with neurons. High-resolution immunofluorescence microscopy analysis confirmed that higher numbers of TIGR4 bacteria bound to neurons and SH-SY5Y cells than the non-piliated TIGR4Δ*rrgA-srtD* (Figs 2A, 2B and S6A, S6B). To study if the pilin adhesin RrgA influences the adhesion to neurons, we used a mutant lacking the tip protein RrgA, TIGR4Δ*rrgA* [9]. We found that adhesion to neurons was significantly reduced in the mutant as compared to wt TIGR4 (Figs 1A and S5A), and reached similar levels as using the pilus-1 mutant TIGR4Δ*rrgA-srtD* (Figs 1A and S5A). Bacterial adherence is a prerequisite for subsequent invasion of host cells. Internalization assays using both neurons and undifferentiated SH-SY5Y cells showed that the mutants TIGR4Δ*rrgA-srtD*, and TIGR4Δ*rrgA*, but not the complemented strain TIGR4Δ*rrgA +rrgA* [12], could invade significantly less than wt TIGR4 (Figs 1B and S5B). Thus, RrgA is the structural component of pilus-1 that mediates pneumococcal adhesion and invasion into neurons.

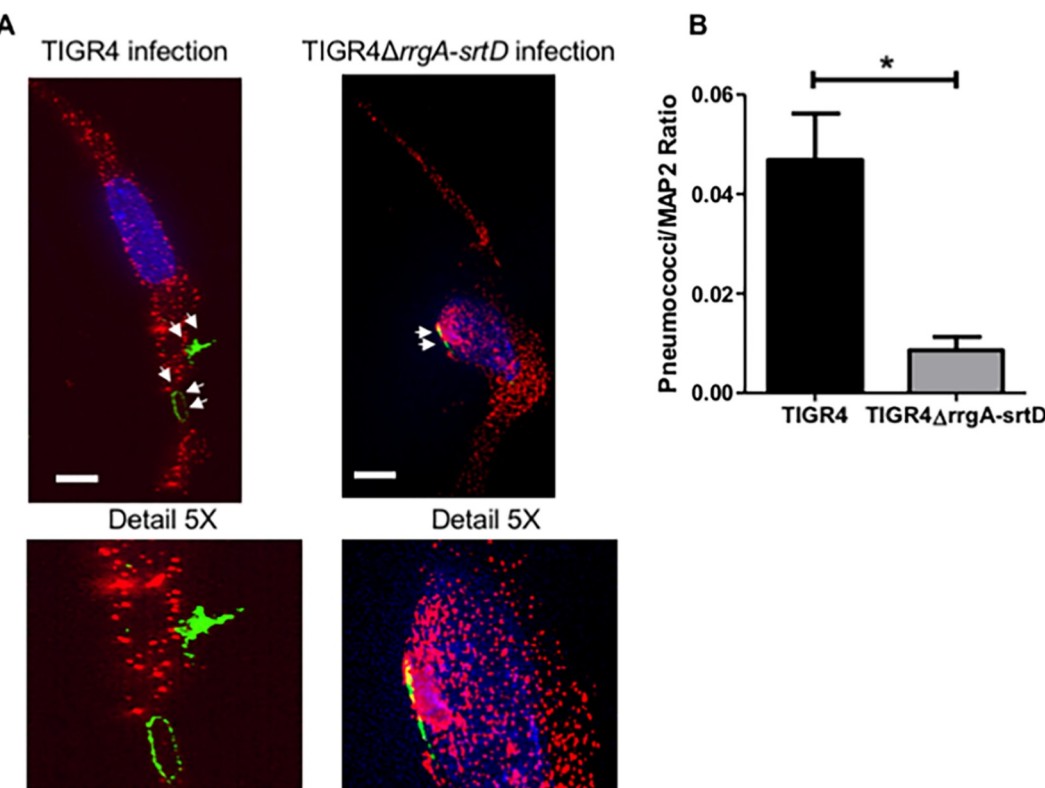

**Fig 2. High-resolution fluorescence microscopy analysis supports that pilus-1 expression increases pneumococcal adhesion to neurons.** (**A**) Piliated TIGR4 bacteria or its isogenic mutant in pilus-1 TIGR4Δ*rrgA-srtD* were used to infect neuronal cells. High-resolution fluorescence microscopy analysis was performed on adhered bacteria where neurons were stained with anti-MAP2 antibody combined with goat anti mouse Alexa Fluor 594 (red), and the pneumococcal capsule stained with anti-serotype 4 capsule antibody combined with goat anti rabbit Alexa Fluor 488 (green). White arrows point to pneumococci that adhered to neurons. White scale bars represent 10 μm. Two representative images are shown selected from 200 cells with adhered bacteria imaged per pneumococcal strain. The panel "Detail 5X" displays a 5X-magnified image of the area in the original images where the bacteria adhered to neurons. (**B**) Quantification analysis of the amount of pneumococcal signal detected on the plasma membrane of neurons by high-resolution microscopy images shown in Fig 3A. For quantification of the bacterial fluorescence signal on neurons, in each image (n = 200 neurons with adhered bacteria per each pneumococcal strain) the area occupied by the green fluorescence signal of the bacteria was divided by the area occupied by the red fluorescence signal of neurons. All areas were measured in square pixels and calculated with the software Image J. Columns in the graph represent average values. Error bars represent standard deviations. The Pneumococci/MAP2 ratio is shown on the Y axis; $^{*}$ = p<0.05.

## The cytotoxin pneumolysin enhances pneumococcal internalization into neurons

In contrast to the mutant strain lacking RrgA, the mutant strain lacking Ply, TIGR4Δ*ply*, adhered to neurons and SH-SY5Y cells to a similar extent as wt TIGR4 (Figs 1C and S5C). However, absence of Ply expression resulted in a significantly lower invasion of neurons and SH-SY5Y cells (Figs 1D and S5D), suggesting that Ply, located on the neuronal plasma membrane, can promote bacterial entry of neurons.

## Both RrgA and Ply interact with neuronal β-actin

To identify proteins on the plasma membrane of neurons that bind to RrgA and/or Ply, we setup a pull-down assay using His-tagged RrgA and Ply coupled to Ni-NTA magnetic beads incubated with a cell lysate of differentiated human neurons, and performed mass spectrometry analysis. The quality of the neuronal cell lysate was assessed by SDS-page and Coomassie

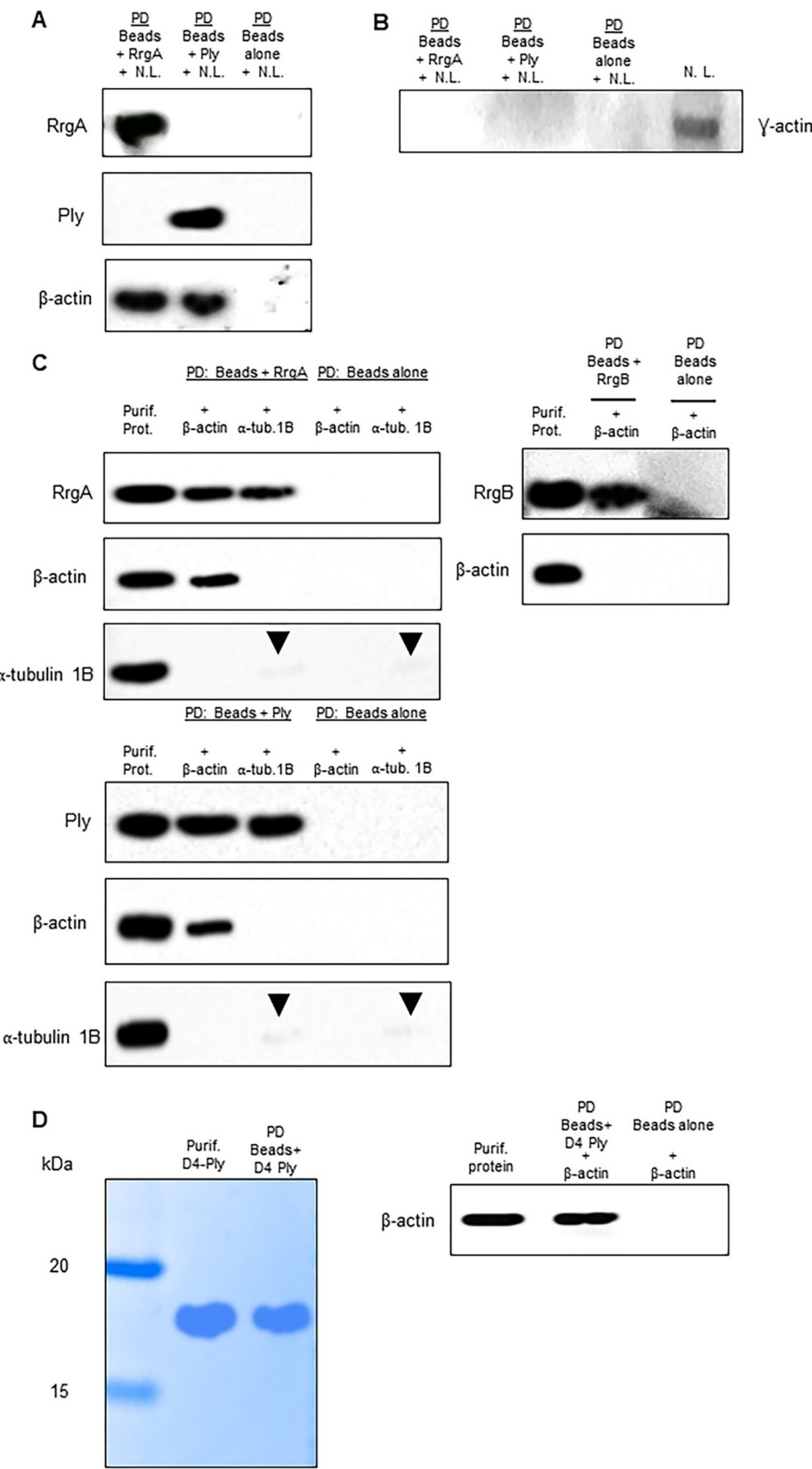

**Fig 3. Pull-down experiments show that RrgA and Ply interact with β-actin of neurons.** (**A**) Western blot analysis performed using the protein content obtained after the pull-down experiments (P.D.) using Ni-NTA beads coupled with purified RrgA or Ply and incubated with neuronal lysate (N.L.). Three separate western blots using the same samples were performed for the detection of RrgA, Ply and β-actin respectively. (**B**) The same P.D. samples used in 4A were also used to assess binding of RrgA and Ply to γ-actin of neurons (**C**) Western blot analysis performed after co-immunoprecipitation experiments using purified RrgA and Ply, coupled with Ni-NTA beads, incubated with the purified recombinant proteins β-actin or α-tubulin 1B (as negative control). Black arrows point to faint bands that were detected for α-tubulin 1B. The presence of these bands of similar intensity in the IP samples with beads coupled with RrgA or Ply and beads alone indicates a slight unspecific affinity of α-tubulin 1B for the Ni-NTA beads. Four separate western blots using the same samples were performed for detection of RrgA, Ply, β-actin and α-tubulin 1B. As a specificity control, western blot analysis was also performed after the co-immunoprecipitation experiments using purified RrgB incubated with the purified recombinant proteins β-actin. Two separate western blots using the same samples were performed for detection of RrgB and β-actin. (**D**) Detection of D4-Ply and Ni-NTA beads coupled with D4-Ply by Coomassie staining, and western blot analysis performed after co-immunoprecipitation experiment using purified D4-Ply, coupled with Ni-NTA beads, incubated with the purified recombinant proteins β-actin.

staining (S7 Fig). We found that the cytoskeleton protein β-actin was the only abundant neuronal protein that bound to RrgA and Ply that was not present in the negative control (Fig 3A and S1–S3 Tables). α tubulin proteins also bound to RrgA and Ply at high scores, but all of them were present in the negative control (S3 Table and S9 Fig). Another actin isoform that is expressed by neurons is γ-actin [32]. To study possible interactions with γ-actin, western blot analysis was performed using the same pull-down samples with neuronal lysate bound to His-tagged RrgA and Ply. Neither RrgA nor Ply could bind to γ-actin expressed by neurons (Fig 3B). To further confirm the specificity of the β-actin interaction, we performed pull-down experiments using recombinant β-actin protein. Western blot analysis showed that both the recombinant RrgA and Ply, but not the pilus-1 backbone protein RrgB, bound to β-actin (Fig 3C). To assess whether RrgA and Ply were interacting exclusively with β-actin and not with other cytoskeleton proteins, we made use of the α-tubulin-1B chain. The pull-down experiment showed that neither RrgA nor Ply bind to α-tubulin-1B, as only a faint α-tubulin-1B band was detected that also appeared in the control with only Ni-NTA beads, indicating an unspecific affinity of α-tubulin-1B for the beads (Fig 4C). Taken together our results suggest that both RrgA and Ply bind specifically to neuronal β-actin.

Since neither of the two receptors pIgR or PECAM-1 were detected in the pull-down assay using mass spectrometry, we also assessed the expression of the two receptors in neurons. Western blot analysis showed that human neurons do not express pIgR, nor PECAM-1 (S8A Fig). This was also confirmed using immunofluorescence microscopy analysis of mouse brain tissues showing that neurons from mice do not express pIgR or PECAM-1 (S8B and S8C Fig).

## β-actin is expressed on the neuronal plasma membrane and its outer surface

We next studied the expression of β-actin on the plasma membrane of neurons. β-actin filaments can reach the plasma membrane of eukaryotic cells as previously reported [33–35]. Furthermore, it was previously reported that motile cells, like neurons, have the peculiarity of having a cytoskeleton with a high plasticity [36]. As a result of such plasticity, some of the actin cytoskeleton filaments that are within the plasma membrane can protrude on the outer space forming so-called patches [37]. To assess the expression of β-actin on the apical side of the plasma membrane of neurons, we performed immunofluorescence microscopy analysis combined with 3D orthogonal views imaging using non-permeabilized neurons to visualize the lipid layer of the plasma membrane, stained with nile red, and β-actin. Notably, both on XZ and YZ orthogonal views the β-actin fluorescent signal was detected above the red signal of the plasma membrane lipid layer (Fig 4A). The same expression above the lipid layer was observed

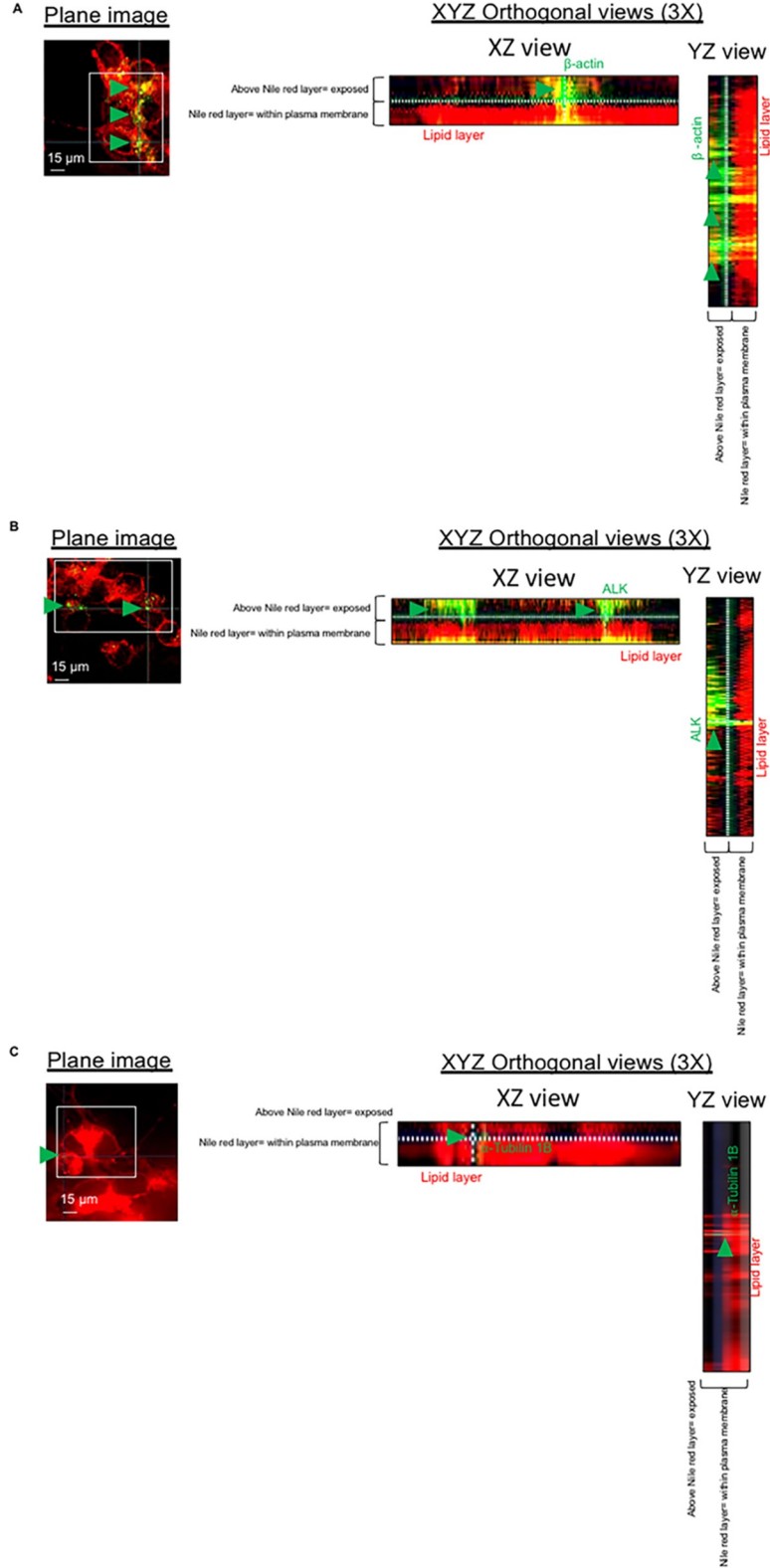

**Fig 4. Exposure of β-actin on the plasma membrane of neurons.** Immunofluorescence microscopy analysis showing fixed and non-permeabilized neurons and stained for nile red for the detection of plasma membrane lipids; neurons were co-stained with anti-β-actin antibody (**A**), anti-ALK antibody (**B**) and anti-α-tubulin 1B antibody (**C**). 3D-orthogonal views were used to assess whether the detection of β-actin antibody (A), anti-ALK antibody (B) and anti-α-

tubulin 1B (C) was above the lipid layer of neuronal plasma membrane. Per each experimental group, n = 200 neurons (among two different biological replicates) were imaged.

for anaplastic lymphoma kinase (ALK), a receptor tyrosine kinase expressed on the neuronal plasma membrane [38], chosen as positive control (Fig 4B). In contrast, only a very minor expression of α-tubulin-1B was detected on the neuronal plasma membrane, and XYZ orthogonal views showed that the α-tubulin-1B fluorescent signal was within the red signal of the lipid layer (Fig 4C). These data demonstrate that α-tubulin-1B is not exposed on the outer surface of the neuronal plasma membrane, in contrast to what we found for some of the β-actin filaments.

## Pneumococci expressing RrgA co-localize with β-actin of neurons *ex vivo*

To investigate if RrgA mediates pneumococcal binding to neurons *in vivo*, we made use of our bacteremia-derived meningitis mouse model [1, 9, 16]. Brain tissue sections from mice infected with TIGR4 or its isogenic mutant TIGR4Δ*rrgA* were examined by high-resolution immunofluorescence microscopy and 3D imaging reconstruction. A significantly stronger bacterial fluorescence signal associated with neurons was observed in the brain of mice infected with wt TIGR4 as compared to mice infected with TIGR4Δ*rrgA* (Fig 5A). Furthermore, we found that TIGR4 co-localized with β-actin of neuronal cells, stained using the specific marker MAP2 (Fig 5B), while infection with the mutant TIGR4Δ*rrgA* showed no co-localization between the bacteria and neuronal β-actin. To ensure that the anti-β-actin antibody used to detect neuronal β-actin was not cross-reacting unspecifically with pneumococci, we stained TIGR4 bacteria with anti-β-actin antibody. Through immunofluorescence microscopy we did not see any β-actin signal on TIGR4 pneumococci (S10 Fig). Hence, these *in vivo* data support our *in vitro* data, suggesting that pneumococcal RrgA interacts with β-actin on neurons.

## Pneumococci exploit β-actin filaments to invade neurons

To study pneumococcal internalization into neurons, we used a CFU-based invasion assay. We found that the double mutant TIGR4Δ*rrgA-srtD*Δ*ply*, in contrast to wt TIGR4, was poorly internalized by neurons (Fig 6A). Infected neurons with intracellular pneumococci were fixed, permeabilized and immunofluorescence staining was performed to detect *S. pneumoniae* cells and β-actin. Quantification analyses confirmed that TIGR4Δ*rrgA-srtD*Δ*ply* bacteria were found only rarely inside neurons (Fig 6B). High-resolution fluorescence microscopy combined with 3D orthogonal view analysis demonstrated that intracellular TIGR4 pneumococci co-localized with β-actin staining (Fig 6C). These results argue that pneumococci, once bound to β-actin on the plasma membrane, retain this binding after becoming internalized, and suggest that pneumococci exploit the cytoskeleton β-actin filaments to enter neurons. The β-actin staining in close proximity to intracellular bacteria was much more intense than the β-actin staining in the rest of the neuronal cell (Fig 6D). Such a higher intensity in that specific portion of the neuronal cell could imply a local increase of β-actin, suggesting that pneumococcal interaction with β-actin on the neuronal plasma membrane could trigger actin polymerization, promoting bacterial invasion and pathogen internalization. In contrast, the very few intracellular TIGR4Δ*rrgA-srtD*Δ*ply* bacteria observed did not colocalize with neuronal β-actin and no increased β-actin fluorescent signals were observed in the zones nearby the bacteria (Fig 6E). However, in other areas of the displayed neuron there were zones in which the β-actin fluorescent signal was more intense (Fig 6E). The zones of intense β-actin fluorescent signal were far from the intracellular bacteria indicating that the increased fluorescent signal could be due to

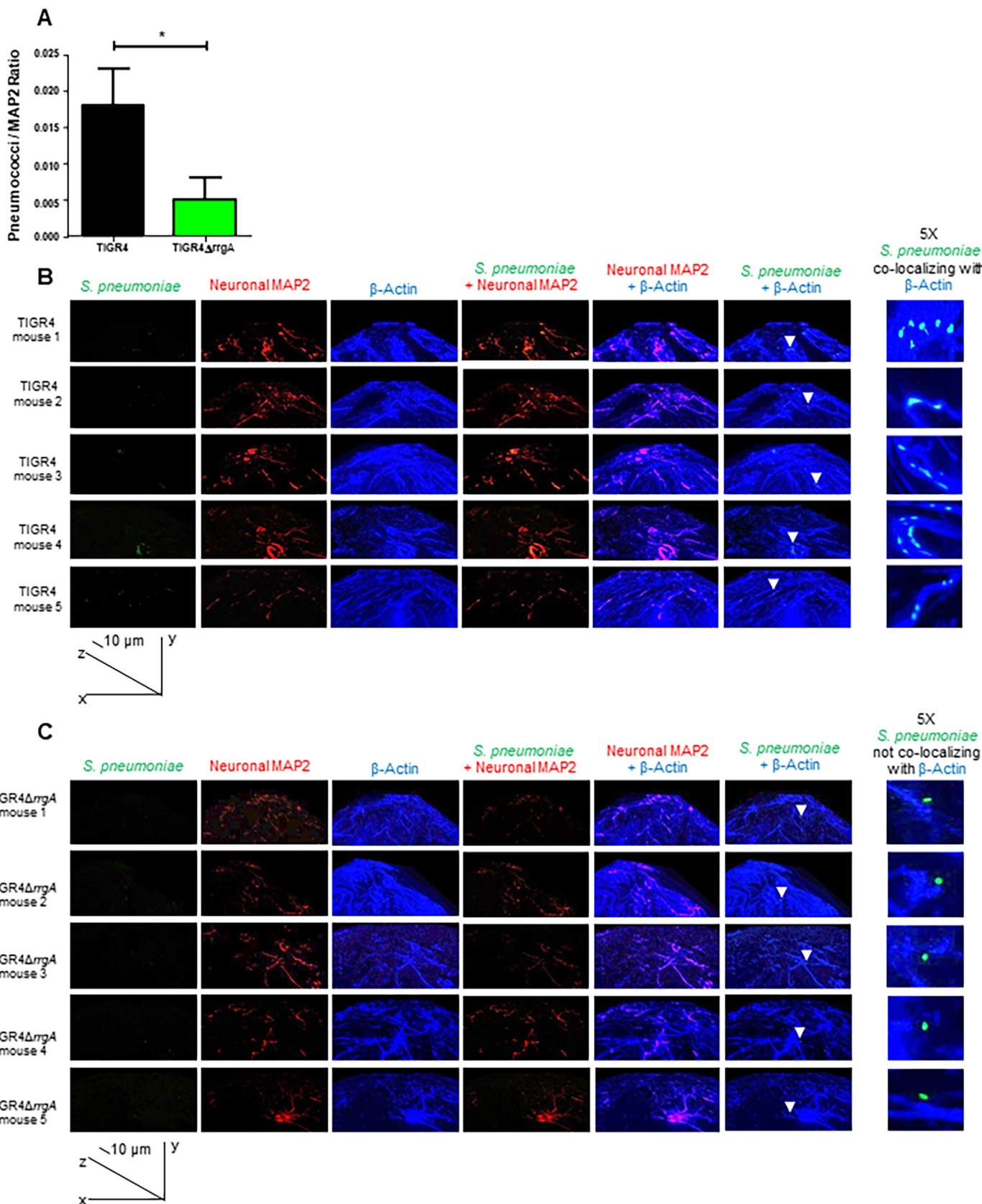

**Fig 5. Mouse brain tissue *ex vivo* analyzed by high-resolution fluorescence microscopy and 3D reconstruction imaging showing that pneumococci associated with neurons co-localize with neuronal β-actin only when expressing RrgA.** Pneumococci were stained with anti-serotype 4 capsule antibody combined with goat anti rabbit Alexa Fluor 488 (green), neurons were stained with anti-MAP2 antibody labelled with Zenon labeling mouse IgG 594 fluorophore (red), β-actin was stained with anti-β-actin antibody combined with goat anti mouse Alexa Fluor 647 (far red, a blue color was assigned using the Softworx imaging software). High-resolution microscopy analysis and 3D reconstruction imaging (Volume viewer function of the Softworx imaging software) was performed to firstly detect pneumococci in the brain tissue of the mice co-localized with neurons, then to analyze the co-localization between pneumococci and β-actin, and analysis of β-actin with the neuronal marker MAP-2 to distinguish β-actin of neurons co-localization. (A) For quantifying the bacterial fluorescence signal on neurons, in each image the area occupied by the green fluorescence signal of the bacteria was divided by the area occupied by the red fluorescence signal of neurons, all areas were measured in square pixels and calculated with the software Image J; columns in the graph represent average values, error bars represent standard deviations, the Pneumococci/MAP2 ratio is shown on the Y axis; * = p<0.05. (B and C) At the bottom left corner of figures in panels B and C the graph shows the angle of the 3D reconstruction of each image (XYZ axes) and the scale bars; six tissue sections for each mouse (5 mice) infected with either TIGR4 (A) or TIGR4Δ*rrgA* (B) were imaged, and per each section twenty-five images in random regions of the section were taken. White arrows in the panel "*S. pneumoniae* + β-actin" point towards specific area of the tissue sections to highlight the co-localization between piliated pneumococci and β-actin (A), and the absence of co-localization between non-piliated pneumococci and β-actin (B); the panel "5X *S. pneumoniae* co-localizing with β-actin" displays the region of brain tissue in close proximity of the white arrows with an enhanced 5X magnification.

Ply separated from the bacteria. Our findings suggest that Ply of pneumococci could cause β-actin polymerization with consequent formation of β-actin aggregates which increase the susceptibility to cell death by apoptosis in eukaryotic cells [39]. The formation of β-actin aggregates by Ply together with the binding of pneumococci to β-actin through RrgA will possibly affect the stability of the cytoskeleton on long term.

## The interaction of Ply with the neuronal plasma membrane enhances RrgA binding to β-actin

Ply binds to cholesterol and form pores in the plasma membrane of host cells [40]. Since we found that Ply interacts with β-actin using pull-down experiments, we then investigated if Ply can facilitate RrgA-mediated pneumococcal adhesion to neurons. Using immunofluorescence microscopy analysis, we found that purified RrgA co-localized with the exposed β-actin on the neuronal plasma membrane, as shown by our immunofluorescence microscopy analysis (Fig 7A), confirming the direct interaction between RrgA and β-actin as we previously showed by pull-down experiments (Fig 3A–3C). We then pre-treated neurons with purified Ply, and then added RrgA, and observed a dramatic increase of the β-actin signal on the neuronal plasma membrane and a significantly higher amount of RrgA attached to the neurons that co-localized with β-actin (Fig 7B).

To assess whether the increased amount of β-actin on the plasma membrane was due to the pore-forming action of Ply, we treated neurons with the cholesterol-binding domain 4 (D4) of Ply, previously described to not form pores unless other parts of the Ply protein are present [41]. Notably, immunofluorescence microscopy analysis showed that also treatment of the neurons with purified D4 increased the amount of β-actin exposed on the plasma membrane, leading to increased levels of attached RrgA, to similar levels as was observed for Ply-treated neurons (Fig 7C). Since it was previously reported that D4 anchorage into the host cell plasma membrane causes alteration of the lipid layer [41], we next performed immunofluorescence microscopy analysis of the lipid layer of the neuronal plasma membrane combined with 3D orthogonal views imaging. We found that D4 treatment leads to more β-actin above the lipid layer of the plasma membrane, similar to what was observed in neurons treated with purified Ply (Fig 8A–8C). The increase was also confirmed by quantification analysis of the expression levels of β-actin above the nile red staining based on the total nile red staining of the neuronal plasma membrane in both XZ and YZ orthogonal views (Fig 8D and 8E). As a control, we then used the Ply mutant strain TIGR4Δ*ply* and found that it also adhered more to neurons when neuronal cells were pre-treated with D4 (Fig 8F). This further indicates that the D4 interaction with the membrane lipid layer may increase the exposure of β-actin towards the outer

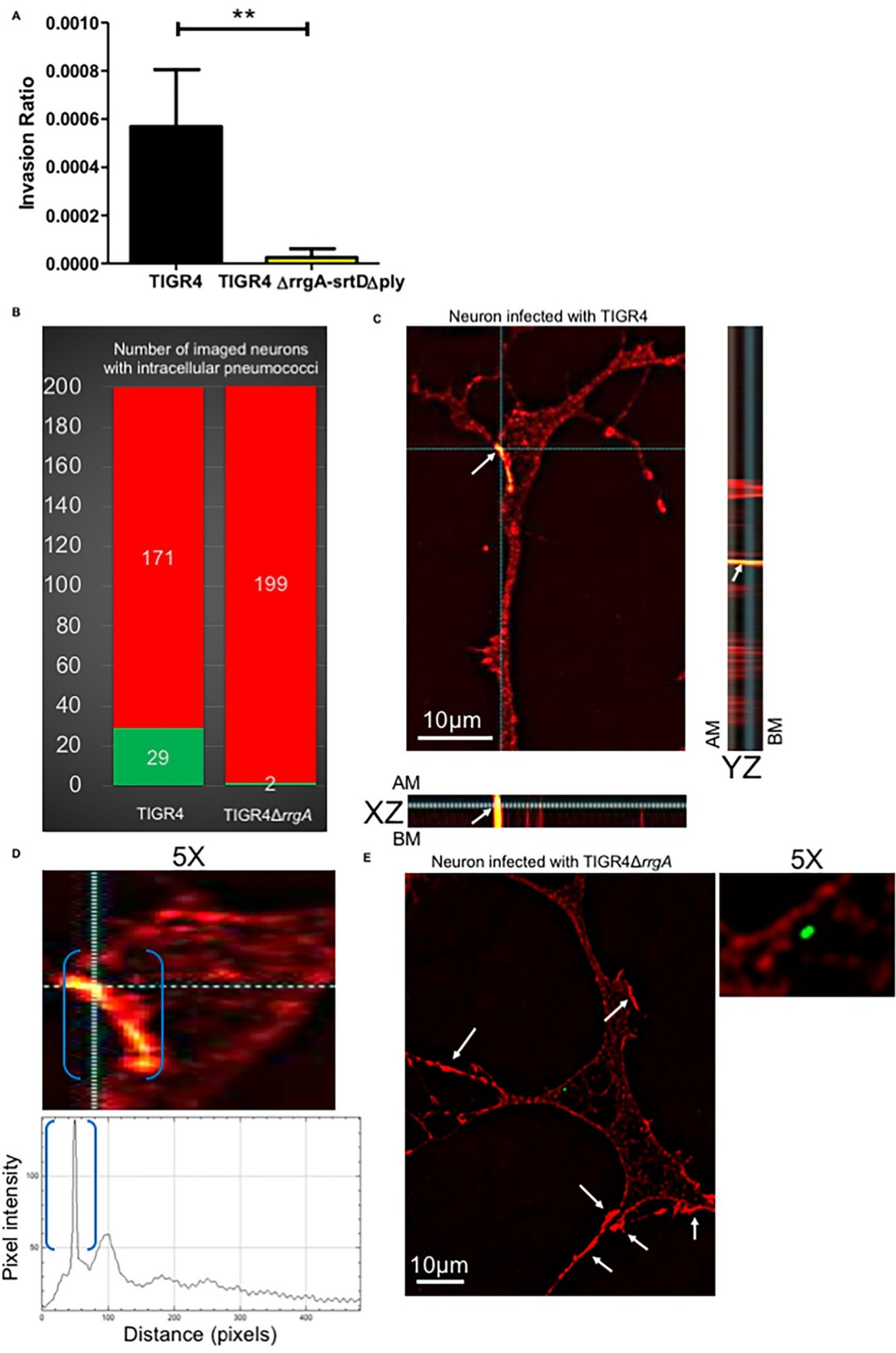

**Fig 6. RrgA and Ply increase pneumococcal invasion of neurons and intracellular piliated pneumococci co-localize with neuronal β-actin.** (**A**) CFU-based internalization assay using wt TIGR4 and its isogenic double mutant TIGR4Δ*rrgA-srtD*Δ*ply*. The uptake ratio by neurons was calculated as [CFU of intracellular bacteria] / [CFU of adhered bacteria)]. In the graph, columns represent average values, and error bars represent standard deviations. The graph shows an overview of three biological replicates. ** = p<0.001. (**B**) High-resolution fluorescence microscopy analysis was performed of neurons with intracellular pneumococci that were fixed, and permeabilized. Immunofluorescence staining was performed to detect β-actin with an anti-β-actin antibody combined with goat anti mouse Alexa Fluor 594 (red) and pneumococci with anti-serotype 4 capsule antibody combined with goat anti rabbit 488 (green). The graph shows a quantification of the number of neurons with intracellular pneumococci among a total number of 200 random neurons imaged per strain, either TIGR4 or TIGR4Δ*rrgA*. The green column represents the number of neurons with intracellular pneumococci and the red column the number of neurons without intracellular pneumococci. (**C**) Neurons with intracellular pneumococci after TIGR4 infection were imaged with z-stacks to capture the thickness of the neuronal cell (number z-stacks = 22). Intracellular pneumococci with the z-stack number = 9 was displayed from top view and in XZ-axes- and YZ-axes-orthogonal views to demonstrate intracellular localization of pneumococci (green). The imaged bacteria were within the neuronal cell thickness between the AM (apical membrane) and BM (basolateral membrane). Both XZ and YZ-axes-orthogonal views showed co-localization between pneumococci (white arrows) and intracellular β-actin. The image shown is a representative of 200 neurons imaged after TIGR4 infection. (**D**) 5X magnification of the neuron shown in Fig 6B focusing on the cell area in close proximity to intracellular pneumococci. The function Profile Plot of Image J was used to measure the intensity (pixels) of the red fluorescence signal of β-actin. Within blue brackets the β-actin staining in close proximity to intracellular pneumococci that corresponds to the pick of fluorescence intensity in the graph underneath the microscopy image is shown. (**E**) Neurons with intracellular pneumococci after TIGR4Δ*rrgA* infection were imaged with z-stacks to capture the thickness of the neuronal cell (number z-stacks = 22). The displayed image shows the z-stack number = 10; white arrows point towards regions of the neuronal cell with a localized enhanced β-actin fluorescent signal. The panel "5X" shows the same image 5X magnified focusing on the area of neuronal cell in close proximity to the intracellular bacteria to highlight the absence of co-localization between TIGR4Δ*rrgA* and intracellular β-actin. This image is a representative of 200 neurons imaged after TIGR4Δ*rrgA* infection.

environment and promote adhesion to neurons through an enhanced pilus-1 β-actin interaction that occurs on the neuronal plasma membrane (Fig 8F). Notably, when pneumococci lack both Ply and pilus-1, despite the increased β-actin on neuronal plasma membrane upon treatment with D4, TIGR4Δ*rrgA-srtD*Δ*ply* ability to adhere to neurons is severely impaired (Fig 8F), further straighten the importance of the pilus-1 in pneumococcal adhesion to neurons through β-actin-interaction. As further control, we also observed that when neurons are treated with the non-cholesterol binding and non-pore-forming Ply, the mutant toxoid PdB [15, 42], and infected with pneumococci that have only the pilus-1 to interact with neurons, TIGR4Δ*ply* adheres similarly to PdB-treated and untreated neurons (Fig 8G). These results are in agreement with a previous report [41], where it was shown that D4 leads to an alteration of the lipid layer structure and an enhanced exposure in the plasma membrane of β-actin. In addition, we also observed that, without using bacteria, treatment with either purified D4 or RrgA alone can lead neuronal cell death, while PdB alone does not cause any significant neuronal cytotoxic effect (Fig 8H). We also observed through pull-down experiments that D4 can interact with recombinant β-actin (Fig 3D). These findings suggest that the binding of D4 to the plasma membrane causes alterations that can lead to neuronal cell death, and that D4 might be the domain mediating the interaction of Ply with β-actin. Moreover, our results also point towards the scenario that the interaction of RrgA with exposed β-actin can cause cytoskeleton instability leading to cytoskeleton disruption and consequent cell death.

To further study the role played by the hemolytic activity of Ply, we next treated the neurons the mutant toxoid PdB. We found that treatment with purified PdB did not increase the amount of β-actin exposed and did not influence RrgA adherence to neurons (Fig 7D). The immunofluorescence microscopy data were also supported by quantification analysis of the amount of RrgA that adhered to the plasma membrane in untreated, Ply-treated, D4-treated, and mutant toxoid PdB-treated neurons (Fig 7D). Taken together, these results demonstrate that the cholesterol-binding domain D4 of Ply, but not the pore-forming activity, is important for increasing the exposure β-actin filaments on the outer surface of the plasma membrane of neurons, and this enhanced exposure of β-actin leads to enhanced pneumococcal binding to neurons through RrgA-β-actin interaction.

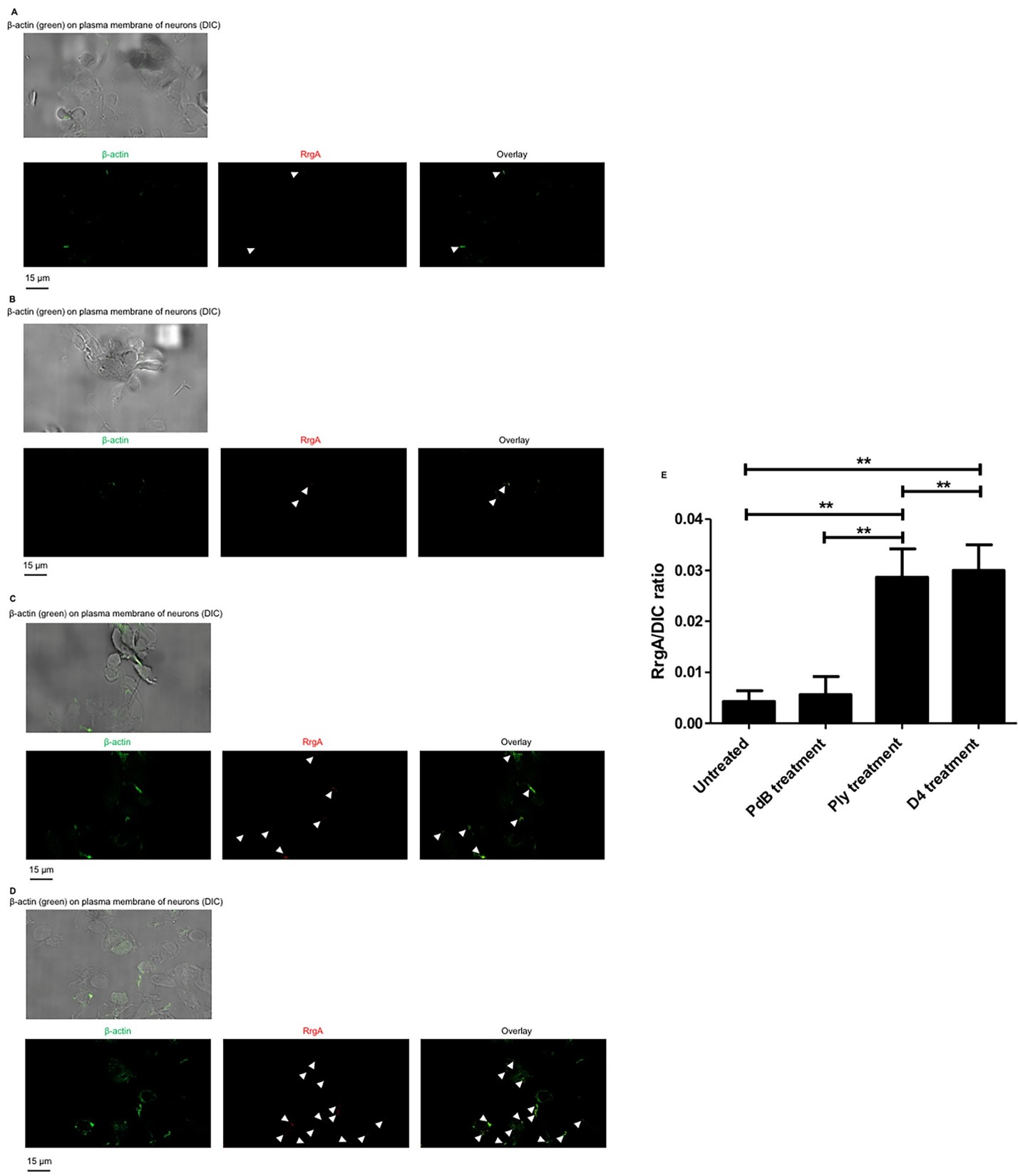

**Fig 7. Ply interaction with the neuronal plasma membrane leads to increased levels of exposed β-actin, allowing more RrgA to adhere to neuronal β-actin.** Immunofluorescence microscopy analysis of fixed and non-permeabilized neurons without any pre-treatment (**A**), treated with non-pore-forming PdB (**B**), full length protein Ply (**C**), or D4-Ply (**D**). In A-D neurons are first shown with the DIC channel to show the neuronal cells seeded on the coverslips, and separate green (FITC-488 nm channel), red (TRITC-594 nm channel) are presented to show the fluorescent signals of β-actin, RrgA, and the overlay panel shows the co-localization of RrgA with β-actin on neuronal plasma membrane. (**E**) For quantification of RrgA fluorescence signal on neurons, in each image (n = 200 neurons with adhered bacteria, per each pneumococcal strain) the area occupied by the green fluorescence signal of the bacteria of RrgA was divided by the total area occupied neurons imaged through the DIC channel. All areas were measured in square pixels and calculated with the software Image J. Columns in the graph represent average values, and error bars represent standard deviations (n = 200 neurons with adhered RrgA imaged, per experimental group). The RrgA / DIC ratio is shown on the Y axis; ** = p<0.01.

## Pneumococcal expression of Ply dramatically enhances intracellular calcium (Ca$^{2+}$) levels of neurons, suggesting disruption of β-actin filaments

The formation of β-actin aggregates by Ply together with the binding of pneumococci to β-actin through RrgA might alter the stability of the neuronal cytoskeleton during a pneumococcal infection. The interplay between the actin cytoskeleton and Ca$^{2+}$ signaling was previously shown to play an important role during actin polymerization and neuronal growth and motility [43, 44]. Moreover, high levels of intracellular Ca$^{2+}$ were described to be cytotoxic in neurons [45–47]. Therefore, we performed Ca$^{2+}$ imaging of neurons infected with wt TIGR4, the double mutant TIGR4Δ*rrgA-srtD*Δ*ply*, the single mutants TIGR4Δ*rrgA* and TIGR4Δ*ply*, and non-infected neurons, using the fluorescent Ca$^{2+}$ indicator Fluo-8. We observed a significant increase of the Ca$^{2+}$ levels in neurons infected with wt TIGR4 compared to neurons infected with the double mutant TIGR4Δ*rrgA-srtD*Δ*ply* or non-infected neurons (S11A–S11C and S11H, S11I Fig). While the level of intracellular Ca$^{2+}$ in neurons infected by TIGR4Δ*rrgA* was only moderately lower (not significant) than with TIGR4 (S11E, S11H and S11I Fig), absence of Ply caused a significant decrease of intracellular Ca$^{2+}$ levels (Figs 8H and 8I and S11D). Furthermore, when neurons were infected with TIGR4Δ*ply* in combination with purified Ply, the levels of intracellular Ca$^{2+}$ reached similar levels to that observed in neurons infected with wt TIGR4 (S11F and S11I Fig), suggesting that the increased Ca$^{2+}$ levels is mainly caused by Ply released by pneumococci. Notably, we also analyzed the Ca$^{2+}$ levels in neurons infected with TIGR4Δ*rrgA* in combination with purified RrgA, and again we observed that intracellular Ca$^{2+}$ levels reached similar levels to what measured in TIGR4-infected neurons (S11G and S11I Fig). An intact actin cytoskeleton was previously described to inhibit the activation of Ca$^{2+}$ entry [48]. Moreover, it has been shown that actin assembly is not mediated by a Ca$^{2+}$ increase but that actin disassembly is accompanied by elevated intracellular calcium levels [49]. Here, we observed that, relative to TIGR4, the average values of intracellular Ca$^{2+}$ peak intensities showed a 50% and 10–15% reduction for neurons infected with TIGR4Δ*ply*, or TIGR4Δ*rrgA* respectively (S11I Fig). This suggests that Ply and to a lesser extent RrgA enhance intracellular Ca$^{2+}$ levels in neurons that likely lead to disruption of β-actin filaments and eventually to neuronal cell death.

## Ply and RrgA promote disruption of β-actin filaments

To verify that Ply and RrgA influence disruption of β-actin filaments, we next investigated neurons infected with TIGR4, TIGR4Δ*rrgA-srtD*Δ*ply*, and non-infected neurons stained for β-actin using STED super-resolution microscopy imaging. Neurons were permeabilized in order to obtain a comprehensive overview of the β-actin filaments within the neuronal cells. We found that β-actin filaments of non-infected neurons showed an intact structure with a continuous β-actin fluorescent signal (Fig 9A), and the double mutant strain, lacking both RrgA and Ply, did not cause any major disruption of the β-actin structures (Fig 9B). In contrast, neurons infected with wild-type pneumococci, expressing both RrgA and Ply, showed a significantly less intense β-actin fluorescent signal, and several gaps of the fluorescent signals were observed

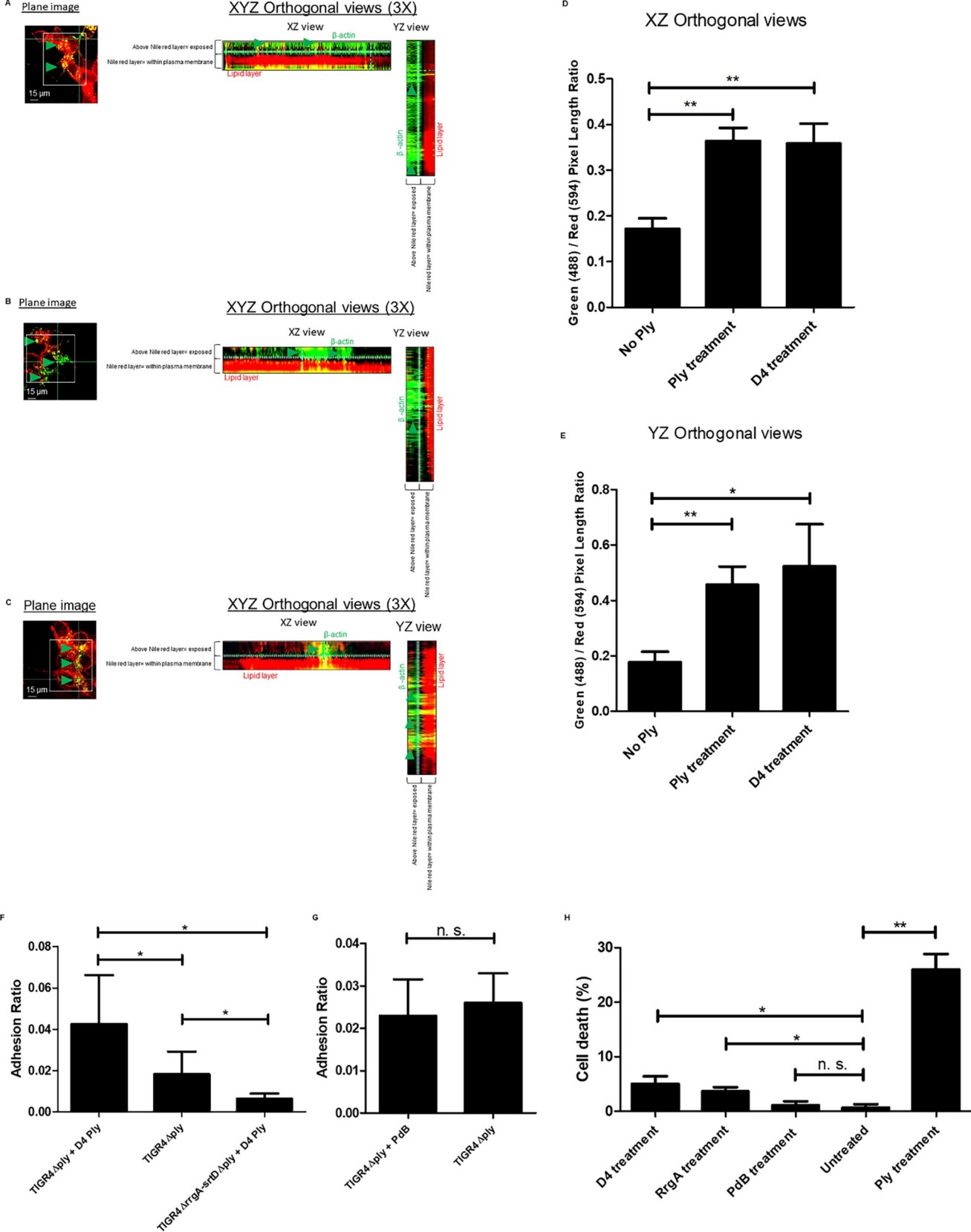

**Fig 8. Treatment with purified Ply and D4 enhance exposure of β-actin on the neuronal plasma membrane.** Immunofluorescence microscopy analysis showing fixed and non-permeabilized neurons pre-treated with purified Ply (**A**), D4 (**B**), or untreated (**C**), same figure shown in Fig 5A), and stained for the detection of plasma membrane lipids (nile red) and β-actin; 3D-orthogonal views were used to visualize the portions of neuronal cells with β-actin signal above the nile red staining (= exposed on plasma membrane); per each experimental group, n = 200 neurons (among two different biological replicates) were imaged. (**D** and **E**) Quantification graphs showing the amount of β-actin signal exposed on the plasma membrane assessed by measuring the length (pixels) of the β-actin signal detected above the nile red staining; Y axis displays Green (488 nm channel) / Red (594 nm channel) ratio that was calculated by dividing the length (pixels) of the β-actin signal (green) detected above the nile red signal (red) by the length of the nile red signal; measurement of the length values (pixels) was performed with the function of Image J *Analyze/Plot Profile*; bars and error bars in the graphs represent average and standard deviation values calculated among 200 neuronal cells imaged for each experimental group, ** = p<0.01, * = p<0.05 (**F**) Adhesion of TIGR4Δ*ply* to untreated and D4-treated neurons and of TIGR4Δ*rrgA-srtD*Δ*ply* to D4-treated neurons was calculated by dividing the total number of bacteria in each well for each pneumococcal strain after pneumococcal infection by the total number of adhered bacteria in each well for each pneumococcal strain. Columns represent average values, and error bars represent standard deviations; the graph shows data from n = 2 biological replicates (with n = 7 and n = 5 technical replicates per each biological replicate), * = p<0.05. (**G**) Adhesion of TIGR4Δ*ply* to untreated and PdB-treated neurons was calculated by dividing the total number of bacteria in each well for each pneumococcal strain after pneumococcal infection by the total number of adhered bacteria in each well for each pneumococcal strain. Columns represent average values, and error bars represent standard deviations; the graph shows data from n = 3 biological replicates (with n = 3 and n = 3 technical replicates per each biological replicate), * = p<0.05. (**H**) Neuronal cell death measured by LDH release in neurons treated with D4, or RrgA, or mutant toxoid PdB, or full Ply, non-treated neurons were used as control; columns represent average values, and error bars represent standard deviations; the graph shows data from n = 2 biological replicates (with n = 3 technical replicate per each biological replicate), ** = p<0.01, * = p<0.05.

along the β-actin filaments, indicating major signs of disruption (Fig 9C). Measurement of the fluorescent intensity along the β-actin filaments also showed similar levels of pixel intensity from β-actin fluorescent signal of non-infected neurons and neurons infected with TIGR4Δ*rrgA-srtD*Δ*ply*, reaching values of pixel intensity of 120 and 100 respectively (Fig 9D), while the β-actin signal measured from neurons infected with TIGR4 was approximately one third of the intensity, reaching a maximum pixel intensity of 35 (Fig 9D). Also, along the analyzed filaments, the pixel intensity values dropped down to 0 because of the presence of gaps of the fluorescent signal, a clear sign of β-actin filament disruption (Fig 9D). These data suggest that presence of Ply and possibly RrgA leads to disruption of the actin filaments in neurons.

## Anti-β-actin antibody inhibits pneumococcal internalization and the cellular toxicity in neurons

To further evaluate the interaction between pneumococci and β-actin, we next used anti-β-actin antibody in order to block the bacterial binding site of β-actin on the neuronal plasma membrane. After one-hour treatment with anti-β-actin antibody, we observed a significant reduction in the adhesion of TIGR4 bacteria to neurons, while adhesion after isotype control treatment was similar as what was observed in neurons without any antibody treatment (Fig 10A). As a control, we found that TIGR4 adhesion was not affected by neuronal treatment with an antibody targeting ALK, a protein expressed on the neuronal plasma membrane [38] (Fig 10B). These data suggest that the anti-β-actin antibody directly interferes with the β-actin binding to pneumococcI through RrgA.

To further explore if anti-β-actin antibody treatment also affect neuronal cell death, we infected treated and non-treated neurons with *S. pneumoniae* TIGR4 and stained with a live/dead dye. Through high-resolution live-cell imaging microscopy, we demonstrate that the neuronal cell death was significantly decreased after anti-β-actin antibody treatment (Fig 10C).

To study if these results are also valid in the human setting, we used primary neurons differentiated from human neuroepithelial stem cells. By using immunofluorescence high-resolution microscopy analysis, we found that β-actin is expressed also on the plasma membrane of human primary neurons (Fig 10D), and that TIGR4 bacteria co-localize with β-actin (Fig 10D). Using one-hour treatment with anti-β-actin antibody, we observed using high-resolution microscopy and quantification analyses that the adherence of TIGR4 pneumococci was significantly impaired (Fig 10D and 10E), in contrast to what was found using mouse IgG isotype control treatment (Fig 10D and 10E). Also, adhesion of TIGR4Δ*rrgA* bacteria to primary

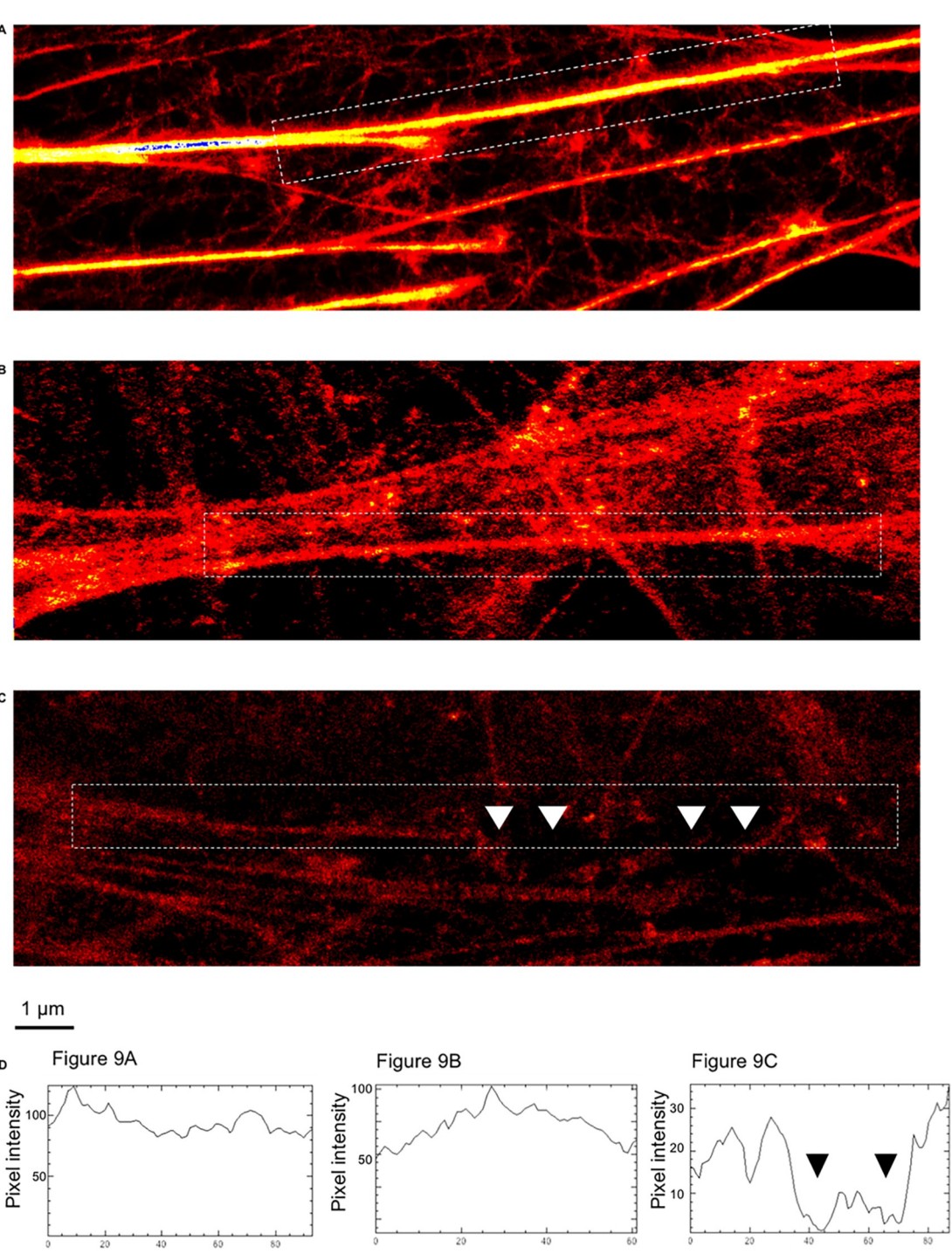

**Fig 9. Disruption of neuronal β-actin filaments imaged by STED super-resolution microscopy.** Immunofluorescence STED super-resolution microscopy analysis showing β-actin filaments (red) within neurons uninfected (**A**), infected with TIGR4Δ*rrgA*-*srtD*Δ*ply* (**B**) and TIGR4 (**C**). (**D**) Intensity profiles of the β-actin filaments selected within white dashed rectangles; fluorescent intensity profile measurement performed with the function of Image J *Analyze/Plot Profile*. the intensity profiles shown in D are representatives of the whole imaging experiment (for each experimental group, 25 neuronal cells were imaged). White arrows in C point towards gaps of β-actin filaments that correspond to the drop of fluorescent intensity measured in the intensity profile graph (black arrows, third graph from the left).

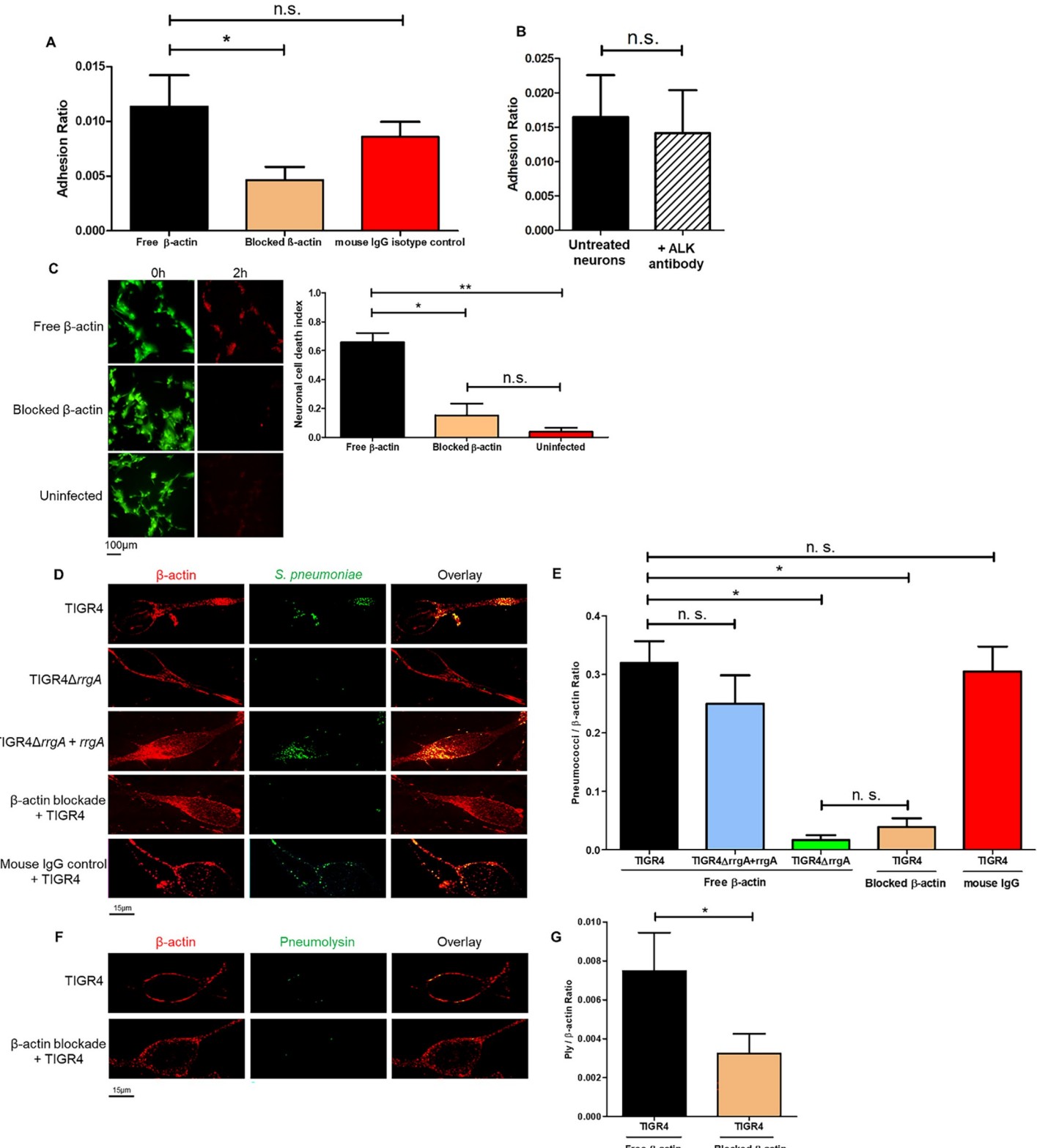

**Fig 10. Blockade of β-actin on the plasma membrane with antibodies inhibit interactions with the cytotoxin Ply.** CFU-based adhesion assays were performed using wt TIGR4 bacteria and in (**A**) untreated neurons, anti-β-actin-antibody-treated neurons and mouse-IgG-treated neurons, and in (**B**) untreated neurons and anti-ALK-antibody-treated neurons. The adhesion ratio was calculated as [CFU of adhered bacteria] / [CFU of (non−adhered bacteria + adhered bacteria)]. The columns represent average values, and error bars represent standard deviations. The graph shows an overview at three biological replicates. $^*$ = p<0.05, n.s. = non-significant. (**C**) Images

from the live-cell imaging experiment at the start (0 h) and at the end, 2 hours post infection, and related quantification of neuronal cell death. Differentiated neurons were stained with a live/dead dye expressing green fluorescence for live cells and red fluorescence when undergoing cell death. Ratio Green (488 nm) / Red (594 nm) represents the neuronal cell death index, calculated by dividing the total area occupied by the green fluorescence signal at time 0 by the total area occupied by the red fluorescence signal at 2 hrs (as performed for data shown in Fig 1). Per each condition with pneumococci, a total of four biological replicates were used, and a total of two biological replicates were used for uninfected neurons. Columns in the graphs represent average values, and error bars represents standard deviations. ** = $p < 0.001$, * = $p < 0.05$, n.s = non-significant. (**D and F**) Infected neurons were fixed and stained with anti-β-actin antibody combined with goat anti mouse Alexa Fluor 594 for the detection of β-actin, and with anti-serotype 4 capsule antibody combined with goat anti rabbit Alexa Fluor 488 (green) for the detection of pneumococci (D). Anti-Ply antibody combined with goat anti rabbit Alexa Fluor 488 (green) was used for detection of Ply (F). Each neuron imaged in D and F is representative of 200 neurons imaged per each pneumococcal strain. (**E and G**) Quantification analysis of the amount of pneumococcal signal detected on the plasma membrane of neurons by high-resolution microscopy. For quantification of the (E) pneumococcal or (G) Ply fluorescence signal on neurons, in each image (n = 200 neurons with adhered bacteria, per each pneumococcal strain) the area occupied by the green fluorescence signal of the bacteria or Ply was divided by the area occupied by the red fluorescence signal of β-actin. All areas were measured in square pixels and calculated with the software Image J. Columns in the graph represent average values, and error bars represent standard deviations. The Pneumococci/ β-actin ratio is shown on the Y axis; * = $p < 0.05$, n.s. = non-significant.

neurons was significantly lower than for wt TIGR4, while the complemented strain TIGR4Δ*rrgA*+*rrgA* showed similar levels as for wt TIGR4 (Fig 10D and 10E). Furthermore, there was no co-localization between TIGR4Δ*rrgA* bacteria and β-actin (Fig 10D and 10E). Lastly, high-resolution microscopy and quantification analysis showed that anti-β-actin antibody treatment significantly reduced the interaction of Ply with the plasma membrane of neurons (Fig 10F and 10G). Since the Ply signal can emanate from the cytosol of pneumococci, or from the bacterial surface or in a released form, often within extracellular vesicles (EVs) derived from pneumococci [50], we cannot state whether the antibody blockade acts directly or indirectly to prevent Ply binding to the plasma membrane.

## Discussion

An acute episode of pneumococcal meningitis is frequently followed by a variety of neurological sequelae usually developing during the first 90 days after infection [5–8]. Symptoms may come from cranial nerve dysfunctions leading to hearing loss and vision impairment, or from motor neurons, leading to hemiplegia or ataxia. But neuronal sequelae may also result in epilepsy, and psychiatric disorders. The distinct types of neurological sequelae associated to pneumococcal meningitis suggest that bacteria may infect neuronal cells in the brain, leading to neuronal cell death, and that the types of sequelae reflect their localization in the brain and what function these missing neurons had [5–8].

In this study we demonstrate that the pneumococcal strain TIGR4 both adheres to and is taken up intracellularly by neurons leading to neuronal cell death. Adherence to neuronal cells was found to be primarily mediated by the pilus-1 adhesin RrgA. Our data further show that purified RrgA can interact with neuronal β-actin. The crystal structure of the pneumococcal pilus-1 associated RrgA adhesin has revealed that its D3-domain exhibits an integrin-like fold [51]. It is known that the actin cytoskeleton interacts with integrins of the extracellular matrix (ECM) and integrin-actin interaction is crucial for cellular adhesion to the ECM and cell-to-cell stability [52]. Furthermore, integrin adhesion complexes possess actin-polymerization activity [53]. Also, we have previously demonstrated that purified RrgA stimulates macrophage motility through interaction with the CR3-integrin [54]. We therefore suggest that RrgA through domain D3 acts as an integrin mimic directly interacting with β-actin, allowing bacterial adhesion to membrane sites on neurons that can respond dynamically via actin polymerization. The direct interaction between domain D3 and β-actin remains though to be shown in future studies. The importance of RrgA for brain infection is also supported by previous studies demonstrating that RrgA promotes bacterial entry into the brain from the blood-stream by interacting with the two endothelial receptors PECAM-1 and pIgR, and at autopsy, after a fatal pneumococcal meningitis, five out of six brains studied contained RrgA expressing pneumococci [16]. However, only about 20–30% of clinical pneumococcal isolates harbour the pilus-1 islet [55, 56]. Thus, more

data are required to be able to conclude whether expression of RrgA increases the risk for developing severe neurological sequelae following an episode of pneumococcal meningitis.

Moreover, we found that pneumococcal invasion of neurons increases by expression of the pilus-1 adhesin RrgA and the cytotoxin Ply which we show both interact with neuronal β-actin. Ply forms transmembrane pores at cholesterol rich sites in eukaryotic membranes, and recently it was demonstrated that domain 4 (D4) of Ply not only binds to cholesterol, but also to the mannose-receptor MRC-1, and that this interaction promotes pneumococcal uptake by MRC-1 expressing dendritic cells and M2-polarized macrophages [15]. We here provided evidence that Ply interacts with the cytoskeleton protein β-actin that is exposed on the neuronal plasma membrane, thereby promoting neuronal cell death likely through β-actin filament disruption and cytoskeleton instability, and aggregate formation with consequent cytoskeleton instability. We also observed, using purified proteins, that Ply and the cholesterol-binding domain 4 of Ply (D4) increase the exposure of β-actin on the plasma membrane, thereby increasing the number of sites that RrgA can bind to, thus leading to enhanced RrgA-β-actin interactions. D4 alone does not cause pore-formation, but has been described to be part of the formation of the pre-pore together with other domains of Ply by penetrating into the host plasma membrane [42] (S12 Fig). Using purified D4 we provide evidence suggesting that the interaction between D4 and the plasma membrane of neurons lead to cell death possibly due to alterations of the lipid layer structure of the membrane (S12 Fig). However, this needs to be studied further in a follow-up study. In neuronal cells, the actin cytoskeleton is connected to cholesterol rich lipid rafts and the dynamic interaction between lipid rafts and the actin cytoskeleton regulates many aspects of neuronal cells [57]. We therefore suggest that Ply may interact with β-actin in neuronal lipid rafts, leading to localized formation of cholesterol-dependent membrane pores, affecting the actin cytoskeletal dynamics at these sites, thereby promoting pneumococcal uptake through endocytosis.

We observed that pneumococci induce neuronal cell death that involves both bacterial adherence via RrgA, and expression of the cytotoxin Ply. Ply is not actively secreted by pneumococci since it lacks a secretion signal sequence, unlike other pore-forming toxins, instead the toxin is released through bacterial autolysis, or as recently suggested, in extracellular membrane vesicles [50]. The main bacterial localization of Ply is in the cytosol, but it was also recently found on the bacterial surface [15]. Exogenously supplied Ply has previously been shown to have neurotoxic effects even in the absence of bacteria [58, 59].

Furthermore, we found that Ply interacts with β-actin and increases $Ca^{2+}$ levels in the neurons. This is in agreement with a previous study showing that purified Ply has neuro-toxic effects that involves calcium influx [58, 60]. For eukaryotic cells, the interplay between the actin cytoskeleton and $Ca^{2+}$ plays a fundamental role for cell growth and motility [22], and actin cytoskeleton proteins have been shown to inhibit $Ca^{2+}$ mobilization from internal stores, a protective cellular response as high levels of $Ca^{2+}$ contribute to neuronal cytotoxicity [45–47]. We observed that increased calcium levels occurred concomitantly with an increased invasion of RrgA-expressing pneumococci into neurons. This suggest that localized calcium, in close proximity to β-actin to which the bacteria adhere, may activate local actin polymerization, resulting in endocytic uptake of pneumococci via a neuronal membrane repair process. This is supported by our finding that endocytosed pneumococci and Ply co-localized with β-actin. We further demonstrate, using STED super-resolution microscopy, that Ply expression leads to disruption of β-actin filaments in the plasma membrane, thus potentially leading to cytoskeleton instability and neuronal death. This is in line with a previous report showing that Ply interaction with the eukaryotic plasma membrane causes remodelling of the actin cytoskeleton [28]. Our study therefore provides evidence that all pneumococci, also the strains that do not express pili, can cause neuronal cell damage through the action of Ply.

In conclusion, we show that pneumococci interact with primary human neurons as well as with differentiated and non-differentiated neurons, and induce neuronal death that is dependent on the presence of the pneumococcal pilus-1 adhesin RrgA and the cytotoxin Ply. Moreover, we find that RrgA promotes binding to neurons, and together with Ply enhance pneumococcal entry into neurons. We also observe that both RrgA and Ply can interact with the cytoskeleton protein β-actin, and this interaction leads to disruption of β-actin filaments eventually leading to neuronal cell death. Using anti-β-actin antibodies we could inhibit pneumococcal invasion and cellular toxicity in the neurons. Since neurons usually do not regenerate, these data suggest that the two proteins and their interaction with β-actin could be important for why pneumococci cause severe neurological sequelae. A more in-depth epidemiological study among patients with pneumococcal meningitis is required to be able to conclude whether the expression of RrgA increases the risk for developing severe neurological sequelae following an episode of pneumococcal meningitis.

## Material and methods

### Ethics statement

All animal experiments were approved by the local ethical committee (Stockholms Norra djurförsöksetiska nämnd).

### Culture of human cells

Neuroblastoma cells (SH-SY5Y, ATCC CRL-2266) were cultivated using Eagle's Minimum Essential Medium (EMEM) and Ham's F12 (Gibco) medium, 15% fetal bovine serum (FBS) (Thermo Fisher Scientific) and 1% Penicillin-Streptomycin (PIS) (Thermo Fisher Scientific) as previously described [29]. SH-SY5Y cells were differentiated using 10 μM Retinoic Acid (RA) (Bio-Techne) for 7 days, 50ng/ml Brain Derived Neuronal Factor (BDNF) (Sigma-Aldrich) was added at day 5 of differentiation [29].

Primary neurons were obtained from the iPS Core Facility at Karolinska Institute. Briefly, neuroepithelial stem (NES) cells were seeded at low (20–30,000 cells/cm$^2$) and high (40–50,000 cells/cm$^2$) density in 24 well plates with glass coverslips, cultivated at 37°C/5% $CO_2$ with DMEM/F-12+Glutamax (Gibco) and 1% PIS, induced with 1% N-2 (Gibco) and 0.1% B-27 (Gibco) supplement in every 2 days for 20 days for differentiation.

### Cultivation of *S. pneumoniae*

Strains used in the study include wild-type TIGR4, the isogenic deletion mutants of pilus-1 (TIGR4Δ*rrgA-srtD*) [9], RrgA (TIGR4Δ*rrgA*) [9] and pneumolysin (TIGR4Δ*ply*) [15]. The double mutant strain T4Δ*rrgA-srt*DΔ*ply* was constructed by replacing the *ply* ORF with a kanamycin resistance gene in the previously constructed non-piliated strain T4Δ*rrgA-srtD* [12]. Briefly, the upstream region of the *ply* gene was amplified by polymerase chain reaction (PCR) using primers *ply-1* and *ply-2*, while the downstream region was amplified with primers *ply-3* and *ply-4*. The kanamycin resistance gene was obtained with primers *kanRfwd* and *kanRrev*. Primers *ply-2* and *ply-3* contain an overhang which overlaps with the *kanRfwd* and *kanRrev* primers, respectively. An overlap PCR reaction using primers *ply-1* and *ply-4* and the three DNA fragments yielded a single PCR product containing the *ply* region with a Kan$^R$ ORF replacement which was then transformed into the T4Δ*rrgA-srt* strain and selected in blood agar plates with kanamycin (200 μg/mL). Deletion was confirmed via sequencing. Primers used are listed in S4 Table. All strains were plated on blood agar overnight at 37°C. Bacterial

colonies were collected and grown in Todd-Hewitt broth (0.5% yeast extract) at 37˚C and harvested at $OD_{600}$ = 0.45–0.5, aliquots stored at -80˚C as a 15% glycerol solution.

## Infection of neurons

SH-SY5Y cells were seeded at a density of 300,000 cells/well in 6 well plates for CFU adhesion assays; 900,000 cells/well in 6 well plates for CFU invasion assay. For Immunofluorescence, cells were seeded at a density of 150,000 cells/well in 12 well plates with a glass coverslip in seeding medium (1:1 EMEM: F12, 5% FBS). Incubated at 37˚C/5% $CO_2$ overnight, cells were then washed with PBS after overnight cultivation, replaced with new seeding medium and incubated for one hour prior to the infection assay.

Neuronal differentiation: SH-SY5Y cells were seeded at a density of 400,000 cells/well in 6 well plates without coverslip and 35,000 cells/well in 12 well plates with a glass coverslip in seeding medium (1:1 EMEM: F12, 5% FBS, 10µM RA). Incubated at 37˚C/5% $CO_2$ overnight, cells were washed with PBS after overnight cultivation, replaced with new seeding medium and incubated for one hour prior to the infection assay.

Primary neurons were differentiated from NES cells in 24 wells with glass coverslips. At day 21 of differentiation, cells were washed once with PBS and replaced with seeding medium (DMEM/F-12, 5% FBS, 1%N-2, 0.1% B-27), then incubated for 1 hour at 37˚C/5% $CO_2$ prior to infection assay.

Adherence and invasion assays: SH-SY5Y cells, SH-SY5Y differentiated neurons and primary neurons were infected with multiplicity of infection (MOI) 10. Infected cells were incubated at 37˚C /5% $CO_2$ for two hours, and Colony Forming Unit (CFU) counting or Immunofluorescence staining was performed. CFU counting: To assess pneumococcal adhesion, after two hours of incubation, supernatant was collected from each well (non-adhered bacteria), wells were washed with PBS to eliminate unbound bacteria. Cells were then treated with 1 ml of Trypsin for 10~15 min, cell suspensions were collected (adhered bacteria). When D4-Ply or PdB-treated neurons were used for adherence assays, neurons were pre-treated with 50ng/mL of D4-Ply or mutant toxoid PdB for 15min, washed once with PBS, and then infected. Bacteria collected prior to and after adhesion were serial diluted in PBS and plated on blood agar at 37˚C/ 5% $CO_2$ overnight. To assess pneumococcal invasion of neurons, after two hours of incubation, wells were washed with PBS to eliminate unbound bacteria and treated with 150 µg Gentamycin (Gibco) in 1 ml of medium for one hour to kill extracellular bacteria. Cells were then washed with PBS and lysed with 1% Saponin (Sigma Aldrich) solution, cell suspensions were diluted with PBS and plated in blood agar at 37˚C /5% $CO_2$ overnight. Adhesion ratio was calculated as [CFU of adhered bacteria] / [CFU of (non−adhered bacteria + adhered bacteria)], uptake ratio was calculated as [CFU of internalized bacteria] / [CFU of adhered bacteria)].

Blockade of β-actin: differentiated neurons and primary neurons were treated with 2µg of either anti β-actin (mouse IgG1) Antibody (Thermo Fisher Scientific), or normal mouse IgG1 (Abcam) as isotype control, or anti-ALK antibody (Thermo Fisher Scientific) as control for the specificity of β-actin blockade, centrifuged with slow rotation at 500 RPM for 5 minutes to enhance the binding of antibodies on the cell surface, then incubated for one hour at 37˚C/5% $CO_2$ prior to infection. Cells (differentiated neurons and primary neurons) were then infected with TIGR4 for 2 hours followed with CFU assay for differentiated neurons, and fixation followed by immunofluorescence staining for primary neurons.

## Mouse experiments

We used the bacteremia-derived meningitis model previously described by our group [1, 9, 16, 61]. Briefly, 5 male C57BL/6 wild-type mice 5 to 6 weeks old (Charles River) per experimental

group were used that were anesthetized by inhalation of isofluorane (Abbott) before infection. 100 μl of 5x10$^7$ CFU were injected intravenously into the tail and the mice were sacrificed at 10 hours post-infection. Clinical symptoms of the infected mice were observed during the infection according to ethical permit. After sacrifice, all mice were perfused to remove all bacteria still present in the blood of brain vessels, perfusion was performed as previously described [1, 9, 16, 61]. Brains were collected, cryopreserved in Shandon Cryomatrix (Thermo Fisher Scientific) and stored at -80˚C.

## Immunofluorescence staining

After two hours of infection, human cells (SH-SY5Y, differentiated neurons, primary neurons) were washed with PBS then fixed with a 4% paraformaldehyde (PFA), permeabilized with 0.1% Triton X-100 in PBS and finally incubated with 1% Bovine serum albumin (BSA) solution in PBS for one hour. The permeabilization step was not performed for cell cultures used to detect bacterial adhesion on plasma membranes. Next, cells were incubated with primary antibodies for one hour and washed with PBS, followed by secondary antibodies for one hour in the dark and washed with PBS. Undifferentiated SH-SY5Y cells were stained with Phalloidin (Thermo Fisher Scientific) for one hour and washed with PBS, finally stained with DAPI for 10 minutes and washed with PBS. All antibodies were prepared using 2% BSA in PBS, 1:50 dilution for primaries, 1:200 for secondaries. Anti-capsule serotype 4 (rabbit) antibodies (Statens Serum Institute, Denmark) were used as primary antibodies for the detection of *S. pneumoniae* and followed with Alexa Fluor 488 goat anti rabbit antibody (Thermo Fisher Scientific). SH-SY5Y cells were stained with Phalloidin 594 (Thermo Fisher Scientific). Anti MAP2 (mouse IgG1) and gamma Enolase Antibodies (NSE-P1, mouse IgG1) (Santa Cruz Biotechnology) were used for staining of differentiated neurons and followed with Alexa Fluor 594 goat anti mouse antibody (Thermo Fisher Scientific). Anti β-actin (mouse IgG1) antibody (Thermo Fisher Scientific) used for staining of primary, differentiated neurons and mouse tissue, followed with Alexa Fluor 594 goat anti mouse antibody (Thermo Fisher Scientific). DAPI (Abcam) was used for all differentiated SH-SY5Y cells for staining process (1:5000). For Nile red staining cells were incubated with first with 100ng/mL of Nile red (Sigma Aldrich) and followed with primary antibodies: Anti β-actin, Anti ALK (Thermo Fisher Scientific) and Anti-α-tubulin (Sigma) without permeabilization, then followed with: Alexa Fluor 488 goat anti mouse, goat anti rabbit and goat anti mouse antibody (Thermo Fisher Scientific) respectively. For the detection of Ply, a mouse anti-Ply antibody (Abcam) was used directly labelled with 488 fluorophore using the Zenon Mouse IgG Labeling Kit (Thermo Fisher Scientific). For detection of co-localization of RrgA and β-actin, cells were either treated with 50ng/mL of purified RrgA or treated with combination of 50ng/mL of purified RrgA and 50ng/mL of Ply (full-length Ply, Ply domain 4-D4 and mutant toxoid PdB) for 15min then stained with Anti β-actin and anti RrgA after fixation, secondary antibodies followed as Alexa Fluor 488 goat anti mouse and Alexa Fluor 594 goat anti rabbit, respectively.

For imaging bacterial adhesion to SH-SY5Y, and differentiated neurons, a total of 500 cells with adhered bacteria were imaged per pneumococcal strain used. For imaging bacterial adhesion to primary neurons, a total of 100 cells with adhered bacteria were imaged per pneumococcal strain used. For imaging bacterial invasion of differentiated neurons, a total of 250 cells with intracellular bacteria were imaged. For imaging bacterial interaction with neurons in mouse brain sections, a total of 25 images per each section per each mouse (6 sections per each mouse, in total five mice per group) were taken. For the experiments in which we blocked β-actin on the plasma membrane of neurons, anti-β-actin antibody was first added onto seeded neurons, the excess of non-bound antibody washed with PBS after one-hour incubation with

the antibody, and neurons were infected with pneumococci; we added again the primary anti-β-actin antibody to be sure to stain and detect all the β-actin exposed on the plasma membrane of neurons, and subsequently the secondary antibody targeting the primary anti-β-actin antibody.

*Ex vivo* mouse brain cryopreserved sections: frozen brains embedded in Cryomatrix (Thermo Fisher Scientific) were cut with a cryostat and 3 sections of 20 μm were placed on each microscope glass slide (VWR). Sections were fixed with acetone for 10 minutes, dried and incubated with anti-capsule serotype 4 antibody (rabbit IgG) mixed with anti-MAP2 antibody (mouse IgG) overnight at 4˚C. Slides were washed with PBS and incubated for four hours with an Alexa Fluor 488 goat anti rabbit secondary antibody (for detection of *S. pneumoniae*) mixed with Alexa Fluor 647 goat anti mouse secondary antibody (for the detection of neuronal MAP2). Slides were washed with PBS and incubated for four hours with anti β-actin antibody labelled with 594 fluorophore using the Zenon Mouse IgG Labeling Kit (Thermo Fisher Scientific). For detection of PECAM-1 and pIgR, blood brain barrier vascular in mouse tissue were stained with PECAM-1 or pIgR (Abcam) overnight 4˚C, followed with Alexa fluor 488 goat anti rat or rabbit and Lectin for 2 hours at room temperature; Neurons in mouse tissue were stained with MAP2 and PECAM-1 or pIgR overnight 4˚C, followed with Alexa fluor 594 goat anti mouse and Alexa fluor 488 goat anti rat or rabbit for 2 hours at room temperature. Dilutions of primary and secondary antibodies were the same as the ones used for the staining of human cells described above. Vectashild (Vector Laboratories) was finally added to each coverslip with fixed cells or stained mouse brain section, covered with a coverslip and analyzed by fluorescence microscopy.

## High-resolution fluorescence microscopy and image processing

The high-resolution fluorescence microscope Delta Vision Elite Imaging System (GE healthcare) was used to analyze all immunofluorescence staining experiments. Images were acquired using a scientific complementary metal-oxide-semiconductor (sCMOS) camera and obtained using three different filters at different wavelengths: 488 nm (green), 594 nm (red) and 647 nm (purple, which we digitally changed into blue to have a better contrast with the green and red signals). Images were finally processed with Softworx imaging program (Applied Precision). For imaging adhered bacteria, human cells (SH-SY5Y, differentiated neurons and primary neurons) were fixed and not permeabilized, only the plasma membrane was imaged with a single snapshot for each filter. For imaging intracellular bacteria in differentiated neurons, or bacterial interaction with neurons in mouse brain sections, series of z-stacks were imaged in order to capture either the thickness of the neurons (in case of imaging differentiated neurons) or the cell morphology of neurons within each brain mouse section. The z-stacks Images (z-stacks) taken with the DV Elite Imaging System was rotated using the *3D Volume Viewer* function of the imaging software Softworx.

## Live-cell imaging and cell viability experiments

Differentiated neurons were treated with 2μM Calcein AM and 4 μM Ethidium bromide from LIVE/DEAD Viability/Cytotoxicity Kit for 20 min at 37˚C/5% $CO_2$. Then cells were infected with different strains of TIGR4 at MOI = 10 under a 20X objective microscope of the Delta Vision Elite Imaging System at 37˚C/5% $CO_2$, time lapse images were taken at every 10 seconds for total two hours with green, red and bright field channel.

For further confirmation, cytotoxicity was also determined by the level of enzyme lactate dehydrogenase (LDH) in the supernatant during 2 hours of infection or treatment with 50ng/mL of purified D4 or RrgA following the manufacturer's instructions using Cytotoxicity kit (Roche).

## Ca$^{2+}$ imaging of neurons

Ca$^{2+}$ imaging of differentiated neurons was performed following the instructions of previously described set-up and manufacturer using Fluo-8 (AAT Bioquest) calcium indicator. Briefly, differentiated neurons were treated with 2μM Fluo-8 AM for 30 min at 37˚C/5% CO$_2$, then cells were washed once with medium and replaced with new medium without Fluo-8. Imaging of calcium was performed under a 20X objective microscope (Delta Vision Elite) using FITC fluorescence channel. Cells were infected with or without wt TIGR4 or its isogenic mutants at MOI = 10 for two hours and then Fluo-8 AM was loaded for recording of calcium flux. In order to assess the effect of Ply and RrgA on calcium flux, 100ng/mL of purified Ply or RrgA was added to the cells (infected with TIGR4Δply or TIGR4ΔRrgA for two hours and loaded with Fluo-8 for 30 min) 2 min before imaging. ImageJ was used for measurement of fluorescence intensity from manually chosen region of interest (ROI), intracellular Ca$^{2+}$ levels were determined using $\Delta F/F_0$ as previously described [62]. Originlab was used for background subtraction for all the calcium images.

## STED imaging

STED imaging was performed with an instrument (Abberior Instrument, Göttingen, Germany), built on a stand from Olympus (IX 83), with a four-mirror beam scanner (Quad scanner, Abberior instruments). The excitation beam from a fiber-coupled pulsed diode laser (594 nm, 40 MHz repetition rate, Abberior Instruments) is spatially overlapped with the depletion beam of a pulsed fiber laser (MPB, Canada, model PFL-P-30-775-B1R, 775 nm, 40 MHz repetition rate, 1.2 ns pulse width and 30 nJ pulse energy), reshaped into a donut profile by a phase plate (VPP-1c, RPC Photonics). The two beams are focused onto the sample by an oil immersion objective (Olympus, UPLSAPO 100XO, NA 1.4). The fluorescence is collected through the same objective, passed through a dichroic mirror, a motorized confocal pinhole (MPH16, Thorlabs, set at 50 μm diameter) in the image plane, is then split by a second dichroic mirror and detected by a single-photon counting detector (Excelitas Technologies, SPCM-AQRH-13), equipped with an emission filter (FF01-615/20, Semrock) and an IR-filter (FF01-775/SP-25, Semrock) to suppress scattered light from the depletion laser. Laser triggering, detector gating and image acquisition are controlled by an FPGA card via the Imspector software (Abberior instruments). A spatial resolution (FWHM) of about 60 nm could be reached in this study. The maximum STED power applied was adjusted to obtain images with the least photobleaching while resolving the actin filaments and their morphology. A Gaussian smoothing filter (inbuilt smoothing function in Imspector) was applied on the STED images to reduce the noise. The imaging conditions for non-infected neurons (10A) were somewhat different from the other two groups (10B and 10C), in fact the pixel integration time (pixel dwell time) was 10 microseconds for non-infected while it was 50 microseconds for the infected groups. This was because of the large difference in intensity between the non-infected and the other groups and the detector limitations at very high intensities.

## Quantification of fluorescent signal and co-localization analysis

Fluorescent signals were quantified using Image J as previously described [16]. Briefly, the fluorescent signal to quantify was first converted into a grayscale image and inside each field of view the area covered by the fluorescent signal was selected using the function *Image-Adjust-Color Threshold*. The area covered by each fluorescent signal was then measured using the function *Analyze-Measure* (area measured in pixel$^2$). For quantification of the fluorescent intensity (Fig 6B), the fluorescent signal was first converted into a grayscale image and using the function *Analyze-Plot Profile* of Image J the graph (pixel distance on the X axis, pixel

intensity on the Y axis) was generated. For co-localization analysis, fluorescent signal from different channels were first converted into grayscale images and *Co-loc* plugin of Image J was used to determine the area of co-localization between the two images (co-localization zones assigned automatically by Image J a white color); the area covered by the white color was then selected using the function *Image-Adjust-Color Threshold* and then measured using the function *Analyze-Measure* (area measured in pixel$^2$), and showed as a percentage (%) of the area of the main fluorescent signal (among the two channels analyzed for co-localization) detected. For the quantification of the β-actin exposed on the plasma membrane of neurons pre-treated with Ply and D4, β-actin and nile red signals were manually selected with the *Rectangle polygon selection* of Image J; measurement of the length values (pixels) was performed with the function of Image J *Analyze/Plot Profile*. For the quantification of the STED fluorescence microscopy images, β-actin filaments were manually selected with the *Rectangle polygon selection* of Image J; fluorescent intensity profile measurement performed with the function of Image J *Analyze/Plot Profile*.

## Pull-down experiments with RrgA, Ply and neuronal lysate or purified β-actin and α-tubulin 1B

Pull-down experiments were performed using Ni-NTA magnetic beads (Thermo Fisher Scientific). First, Ni-NTA beads were equilibrated using a washing buffer containing 300 mM NaCl, 50 mM NaH2PO4, 20 mM imidazole, 1.5 mM MgCl2, 0.1% Triton×100, 3% glycerol and then incubated with 2 μM purified Ply, RrgA or RrgB for 30 min. For pull-down using neuronal cell lysate, 5x10$^6$ differentiated neurons were lysed in lysis buffer containing 150 mM NaCl, 50 mM NaH2PO4, 10 mM imidazole, 1.5 mM MgCl2, 0.1% Triton×100, 0.6% of n-Dodecyl β-D-maltoside, 3% glycerol; 0.9 mM DTT, complete protease inhibitor EDTA-free (Sigma Aldrich), and homogenized by passing through a needle. The lysate was centrifuged at 3500 rpm for 10min at 4˚C. The supernatant was incubated with empty or, Ply or RrgA-conjugated Ni-NTA beads at 4˚C for 2 hrs. Beads were washed four times with washing buffer. The bound proteins were eluted by adding LDS sample buffer (Thermo Fisher Scientific) and incubating for 3 min at 55˚C. Eluted proteins were loaded on SDS-PAGE for Coomassie Blue staining and sent for mass spectrometry analysis at the Science for Life Laboratory at Uppsala, Sweden. Proteins were identified according to the criteria of at least two matching peptides of 95% confidence level per protein. Western blot was done after the pull-down assay to confirm the binding of β-actin to RrgA and Ply. For Ni-NTA pull-down using purified proteins, 2 μM purified β-actin or α-tubulin 1B were incubated with empty or, Ply, D4-Ply, or RrgA-conjugated Ni-NTA beads at 4˚C for 2 hrs. Beads were washed four times with washing buffer (containing 30mM instead 20mM imidazole). Washing steps performed using a buffer containing 300 mM NaCl, 50 mM NaH2PO4, 30 mM imidazole, 1.5 mM MgCl2, 0.1% Triton×100, 3% glycerol, the bound proteins were eluted by adding LDS sample buffer and incubating for 5 min at 95˚C. Eluted proteins were loaded on SDS-PAGE for western blot analysis.

## Mass spectrometry analysis

Gel bands were cut into small (1 mm$^3$ pieces), destained using acetonitrile (ACN), washed and exposed to dithiothreitol (DTT) reduction and iodoacetamide (IAA) alkylation. Thereafter the proteins were digested by sequencing grade modified trypsin at a concentration of 12.5 ng/μL in 25 mM ammonium bicarbonate pH 8 overnight at 37˚C. The peptides were extracted by sonication in 60% ACN and 5% formic acid (FA). Finally, the extracted peptides were completely dried to completion and thereafter desalted using the SPE Pierce C18 Spin Columns (Thermo Fisher Scientific). These columns were activated by 2 × 200 μL of 50%

acetonitrile (CAN) and equilibrated with 2 × 200 μL of 0.5% trifluoroacetic acid (TFA). The tryptic peptides were adsorbed to the media using two repeated cycles of 40 μL sample loading and the column was washed using 3 × 200 μL of 0.5% TFA. Finally, the peptides were eluted in 3 × 50 μL of 70% ACN and dried. Dried peptides were resolved in 40 μL of 0.1% formic acid and further diluted 4 times prior to nano-LC-MS/MS. The nanoLC-MS/MS experiments were performed using a Q Exactive Orbitrap mass spectrometer (Thermo Fisher Scientific) equipped with a nano electrospray ion source. The peptides were separated by C18 reversed phase liquid chromatography using an EASY-nLC 1000 system (Thermo Fisher Scientific). A set-up of pre-column and analytical column was used. The precolumn was a 2 cm EASYcolumn (ID 100 μm, 5 μm particles) (Thermo Fisher Scientific) while the analytical column was a 10 cm EASY-column (ID 75 μm, 3 μm particles, Thermo Fisher Scientific). Peptides were eluted with a 35 min linear gradient from 4% to 100% acetonitrile at 250 nL min-1. The mass spectrometer was operated in positive ion mode acquiring a survey mass spectrum with resolving power 70,000 (full width half maximum), m/z 400–1750 using an automatic gain control (AGC) target of 3×106. The 10 most intense ions were selected for higher-energy collisional dissociation (HCD) fragmentation (25% normalized collision energy) and MS/MS spectra were generated with an AGC target of 5×105 at a resolution of 17,500. The mass spectrometer worked in data-dependent mode. The acquired data RAW- files) were processed by Proteome Discoverer software (Thermo Scientific, version [nr 1.4.1.14]) using the Sequest algorithm towards a combined database containing protein sequences from Homo Sapience (26546 entries) downloaded from Uniprot 2019–06. For the identification of the neuronal proteins bound to RrgA and Ply, we have considered the proteins with a score higher than 50 (Supplementary S1–S3 Tables). All proteins with a score higher than 50 bound to RrgA and Ply were also present in the negative control. The protein β-actin was present at much higher scores among the neuronal proteins bound to RrgA and Ply, with scores of 86.13 and 106.26 respectively (S1 and S2 Tables), compared to the negative control in which β-actin had the lowest score (score = 20.41, S3 Table). The score of 20 is usually the threshold for false positive proteins (20), therefore in our case β-actin can be considered a false positive in the negative control highlighting the finding that β-actin could be a specific ligand of RrgA and Ply.

## Western blot analysis and Coomassie staining

3.5x10⁶ cells (differentiated neurons from SH-SY5Y cells) were harvested and lysed in a RIPA buffer containing protease and phosphatase inhibitors; Detroit and HBMEC cells were directly lysed from a frozen vial after thawing process. Lysed cells were centrifuged for 10 min at 4˚C, supernatants were collected and protein concentration was adjusted to 1μg/μl. Samples were boiled in 95˚C for 10 minutes and loaded into NuPage Novex 4–12% Bis-Tris SDS-PAGE Gels (Thermo Fisher Scientific), electroblotting was performed using the Biorad Trans-Blot Turbo Transfer System. Membranes were first incubated overnight with PBS-T (T = 0,1% Tween) supplemented with 5% milk. After washing the membrane with PBS-T, incubation with primary antibodies (1:2000) was performed for three hours, incubation with secondary antibodies (1:5000) was performed for one hour. Antibodies used for Western blot experiments were diluted in PBS-T supplemented with 1% dry milk. The same primary antibodies used for immunofluorescence staining were used. For data shown in Supplementary S1B Fig, a rabbit polyclonal anti-GAPDH antiserum was used for detection of GAPDH as loading control. Horseradish peroxidase conjugated Goat anti-Mouse IgG (GE Healthcare) diluted 1:5000 was used as secondary antibodies for the detection of Mouse IgGs. Quantification of Ply expression in TIGR4 and TIGR4Δ*rrgA-srtD* (S2 Fig) was performed by dividing the intensity values of Ply bands per the intensity values of GAPDH bands; intensity of bands was measured with Image

J as previously described [63]. For Coomassie staining, after SDS-page electrophoresis, NuPage gels were incubated at room temperature with gentle agitation with SimplyBlue SafeStain (Thermo Fisher Scientific) for two hours, and de-stained until clear visualization of protein bands.

## Statistical analysis

Statistical analyses were performed using Prism 5. For two-group comparisons, the non-parametric Wilcoxon´s rank sum test (also known as Mann-Whitney test) was used. For multiple comparisons (more than two groups), the nonparametric ANOVA test was used to assess the presence of the differences between the groups. The ANOVA test was then combined with the Dunn´s test to make pairwise comparisons.

## Supporting information

**S1 Fig. Differentiation of SH-SY5Y cells into mature neurons.** Generation of mature differentiated neurons from human SH-SY5Y neuroblastoma cells was assessed by the expression of neuronal specific markers MAP2 and NSE via (**A**) immunofluorescence microscopy and (**B**) western blot analysis in which GAPDH was used as loading control. White scale bars in S1A Fig represent 100 μm, in the blot in S1B Fig the same protein concentration of both SH-SY5Y and neurons was loaded into the SDS-page gel. (**C**) Through phase contrast light microscopy analysis, we also observed that differentiated neurons displayed the neuronal typical cell-to-cell connections and axon formation (red arrows), black scale bars represent 100 μm. (TIF)

**S2 Fig. Presence of the pneumococcal pilus-1 and Ply enhance neuronal cytotoxicity.** (**A**) Images from live-cell imaging at the start (time 0) and at the end, 2 hours post-infection. Differentiated neurons stained with LIVE/DEAD dye express green fluorescence when they are alive, and a red fluorescence when undergoing cell death. (**B**) Quantification of the neuronal cell death in the 2-hour infection experiment shown in Fig 1A; Green (488 nm) / Red (594 nm) represents the neuronal cell death index, calculated by dividing the total area occupied by the green fluorescence signal at time 0 by the total area occupied by the red fluorescence signal at the end of the infection. Per each pneumococcal strain, a total of 2 biological replicates (2 wells with neurons, each well seeded in a different day) have been used for the 1-hour experiment, and a total of 2 biological replicates (2 wells with neuron, each well seeded in a different day). Columns in the graphs represent average values, error bars represent standard deviations. ** = $p<0.001$, * = $p<0.05$. (**C**) Percentage average values of neuronal cell death calculated setting the average value of neuronal cell death of TIGR4 to 100%. The percentage average values were calculated using the neuronal cell death index values shown in Fig 1B. (TIF)

**S3 Fig. Neuronal cytotoxicity upon pneumococcal infection measured by LDH assay.** Neuronal cell death measured by analysis of LDH release in neurons infected with TIGR4, TIGR4Δ*rrgA-srtD*, TIGR4Δ*ply*, and TIGR4Δ*rrgA-srtD*Δ*ply*. Columns represent average values of LDH release, error bars represent standard deviation values calculated among three biological replicates (each biological replicated included two technical replicates); * = $p<0.05$. (TIF)

**S4 Fig. Expression of Ply in TIGR4 and TIGR4Δ*rrgA-srtD*.** (**A**) Western blot analysis showing similar expression of Ply in TIGR4 and TIGR4Δ*rrgA-srtD;* for both bacterial strains, the same protein content was loaded in the SDS-page. (**B**) Quantification of Ply expression in TIGR4 and TIGR4Δ*rrgA-srtD* calculated by dividing the intensity of Ply bands per the

intensity of the GAPDH (loading control) bands; band intensity values were measured with Image J.
(TIF)

**S5 Fig. RrgA enhances pneumococcal adherence to SH-SY5Y cells, and RrgA and Ply increase pneumococcal invasion of SH-SY5Y cells.** SH-SY5Y cells were challenged with pneumococci of MOI 10 and after 2 hours (**A**) adhesion to and (**B**) invasion of neuronal cells were measured. Strains used were wt TIGR4 and its isogenic mutants in the pilus, TIGR4Δ*rrgA-srtD*, *rrgA*, TIGR4Δ*rrgA*, and the *rrgA* mutant complemented with *rrgA*. TIGR4Δ*rrgA+rrgA*. (**C**) Adhesion and (**D**) invasion were measured using wt TIGR4 and its isogenic mutant in *ply*, TIGR4Δ*ply*. Adhesion ratio was calculated by dividing the total number of bacteria in each well for each pneumococcal strain after 2 hours infection by the total number of adhered bacteria in each well for each pneumococcal strain. For all graphs (A-D) the columns represent average values, and error bars represent standard deviations. Each graph shows data from at least three (n≥3) biological replicates. *** = p<0.0001, ** = p<0.001, ** = p<0.05, n.s. = not-significant.
(TIF)

**S6 Fig. Imaging of piliated and non-piliated pneumococci that adhered to SH-SY5Y cells.** (**A**) High-resolution fluorescence microscopy was used to visualize piliated and non-piliated pneumococci that adhered to SH-SY5Y cells. Neurons were stained with Phalloidin (red), and TIGR4 and TIGR4Δ*rrgA-srtD* were stained with anti-serotype 4 capsule antibody combined with goat anti rabbit Alexa Fluor 488 (green). White arrows point to pneumococci that adhered to SH-SY5Y cells. White scale bars represent 10 μm. The images shown are two representative images selected among 200 cells with adhered bacteria imaged per pneumococcal strain. The panel "Detail 5X" displays a 5X-magnified image of the area in the original images with bacteria that adhered to neurons. (**B**) Quantification of the number of bacteria that adhered to neurons based on the microscopy analysis results shown in S1A Fig. For quantification, the bacterial fluorescence signal on SH-SY5Y cells, in each image (n = 200 SH-SY5Y cells with adhered bacteria, per each pneumococcal strain) the area occupied by the green fluorescence signal of the bacteria, was divided by the area occupied by the red fluorescence signal of SH-SY5Y cells. All areas were measured in square pixels and calculated with the software Image J. The Pneumococci/Phalloidin ratio is shown on the Y axis. Columns in the graph represent average values, error bars represent standard deviations, * = p<0.05.
(TIF)

**S7 Fig. Coomassie staining of cell lysate of differentiated neurons.** Before performing the co-immunoprecipitation experiments, the quality of the cell lysate of differentiated neurons was assessed by SDS-page electrophoresis and Coomassie staining. The clear detection of the neuronal protein bands ranging from low to high molecular sizes suggested good quality of the cell lysate of HBMEC, Detroit and neurons. The numbers on the left side of the image show the protein molecular weight in kDa.
(TIF)

**S8 Fig. Lack of PECAM-1 and pIgR expression in neurons.** (**A**) Detection of PECAM-1 and pIgR in neurons by western blot analysis; HBMEC were used as positive control for PECAM-1 expression, Detroit were used as positive control for pIgR expression, GAPDH was used as loading control. (**B**) Immunofluorescence microscopy analysis using mouse brain tissue sections showing a co-staining of neurons (in red, stained for the neuronal marker MAP2), and PECAM-1 and pIgR (in green); the overlay and the quantification graph (Y axis shows the % co-localization between PECAM-1/pIgR signals with MAP2 neuronal signal) show that neither PECAM-1 nor pIgR fluorescent signals co-localize with neuronal signal. (**C**)

Immunofluorescence microscopy analysis using mouse brain tissue sections showing a co-staining of brain vasculature (in red) and PECAM-1 and pIgR (in green); the overlay and the quantification graph (Y axis shows the % co-localization between PECAM-1/pIgR signals with vascular endothelium signal) show that PECAM-1 and pIgR fluorescent signals mainly co-localize with the vascular endothelium signal. For quantification graphs in B and C, nine mouse brain tissue sections were analyzed, and ten random images were taken per each section; the area occupied by the green fluorescence signal of PECAM-1 and pIgR, and the area occupied by the red fluorescent signal of vascular endothelium (lectin) and neurons (MAP2) were assessed for co-localization using the *Co-loc* function of *Image J*, and co-localization area were finally measured. was divided by the total area occupied neurons imaged through the DIC channel; columns in the graphs in B and C show average values among all images of all sections imaged per group. All areas were measured in square pixels and calculated with the software Image J. (TIF)

**S9 Fig. Neuronal cytoskeleton proteins identified by mass spectrometry.** A pull-down assay was performed using differentiated neurons from SH-SY5Y cells and Ni-NTA beads-coupled-RrgA (RrgA), Ni-NTA beads-coupled-Ply (Ply) or Ni-NTA beads alone (Negative control) to identify proteins that bound to RrgA, or Ply respectively, as compared to beads alone. Mass spectrometry analysis identified the neuronal cytoskeleton proteins listed on the x axis and their presence is presented as Mass spectrometry scores on the Y axis. The dash black line at score = 20 represents the threshold of false positives. (TIF)

**S10 Fig. Assessment of cross-reactivity of anti-β-actin-antibody on pneumococci.** (**A**) Immunofluorescent microscopy analysis performed using TIGR4 pneumococci showing no β-actin fluorescent signal detected on the surface of the bacteria; (**B**) as positive control we used an anti-capsule serotype 4 antibody and the polysaccharide capsule was clearly detected around the bacterial cells. (TIF)

**S11 Fig. Neurons infected with Ply- and RrgA- expressing pneumococci showed increased levels of intracellular $Ca^{2+}$.** Intracellular $Ca^{2+}$ release and influx in neurons infected with (**A**) TIGR4, (**B**)TIGR4Δ*rrgA-srtD*Δ*ply*, (**C**) non-infected neurons (Control), (**D**) TIGR4Δ*ply* and (**E**) TIGR4Δ*rrgA* were determined using Fluo-8 AM two hours post infection. $Ca^{2+}$ imaging was performed for 6 minutes of time lapse in every 1 second using a FITC fluorescence channel. (**F and G**) Neurons were infected with TIGR4Δ*ply* (F) *or* TIGR4Δ*rrgA* (G), then 100ng/mL of Ply or RrgA was added 2 min prior to imaging and $Ca^{2+}$ imaging was performed for 11 minutes of time lapse in every 1 second using FITC fluorescence channel. In A-G, the fluorescence intensity was measured using Image J from the region of interest (ROI) where a minimum of 10 ROI in each field was selected manually. ΔF/F0 was calculated from Image J-generated data (ΔF = Ft-F0, Ft represents the fluorescent intensity in each given time point and F0 represents the average fluorescent intensity of the resting value). Each graph displays the $Ca^{2+}$ flux from 9 neuronal cells of 3 different experiments shown in 360 and 660 seconds. (**H**) Average ΔF/F0 values of relative peaks (n = 9) during 6 minutes within ROI were shown to compare intracellular $Ca^{2+}$ levels; *** = p<0.0001, ** = p<0.001 n.s. = not-significant. Each data point in the graph represents one peak. (**I**) Percentage of average ΔF/F0 values of TIGR4Δ*rrgA*, TIGR4Δ*ply* and TIGR4Δ*rrgA-srtD*Δ*ply* compared to the ΔF/F0 values showed by neurons infected with wt TIGR4 (set to 100%). Each average value was calculated using the $Ca^{2+}$ flux intensity values shown in Fig 7G (each dot in every group is one intensity value). (TIF)

**S12 Fig. Molecular mechanisms of pneumococcal interaction with neurons.** Schematic figure summarizing the molecular mechanism of *S. pneumoniae* interaction with neurons through released Ply and pilus-1 adhesin RrgA.
(TIF)

**S1 Table. List of proteins bound to Ni-NTA-beads-coupled-RrgA identified by mass spectrometry.** β-actin is shown in bold.
(DOCX)

**S2 Table. List of proteins bound to Ni-NTA beads-coupled-Ply identified by mass spectrometry.** β-actin is shown in bold.
(DOCX)

**S3 Table. List of proteins bound un-specifically to Ni-NTA beads identified by mass spectrometry.** β-actin is shown in bold.
(DOCX)

**S4 Table. Primers used in this study.**
(DOCX)

## Acknowledgments

We thank the iPS core facility at the Karolinska Institutet for isolation and differentiation of primary neurons, and the Mass Spectrometry Facility at Uppsala University for the mass spectrometry analysis. We thank Dr. Fariba Foroogh for the technical help with the cryostat tissue cutting, Dr. Peter Mellroth for providing us with the anti-GAPDH antiserum, Prof. Aras Kadioglu for providing purified PdB, the Protein Production Platform at Nanyang Technological University in Singapore for providing purified Ply and D4-Ply, and Novartis GSK for providing anti-RrgA and -RrgB antibodies. Dr. Shigeaki Kanatani for the help provided during the $Ca^{2+}$ imaging experiments, and Prof. Staffan Normark for the scientific discussions.

## Author Contributions

**Data curation:** Mahebali Tabusi, Sigrun Thorsdottir, Maria Lysandrou, Ana Rita Narciso, Melania Minoia, Chinmaya Venugopal Srambickal, Jerker Widengren, Birgitta Henriques-Normark, Federico Iovino.

**Formal analysis:** Mahebali Tabusi, Sigrun Thorsdottir, Maria Lysandrou, Ana Rita Narciso, Melania Minoia, Chinmaya Venugopal Srambickal, Jerker Widengren, Birgitta Henriques-Normark, Federico Iovino.

**Funding acquisition:** Birgitta Henriques-Normark, Federico Iovino.

**Investigation:** Mahebali Tabusi, Federico Iovino.

**Methodology:** Sigrun Thorsdottir, Maria Lysandrou, Ana Rita Narciso, Melania Minoia, Chinmaya Venugopal Srambickal, Jerker Widengren.

**Project administration:** Mahebali Tabusi, Federico Iovino.

**Resources:** Birgitta Henriques-Normark, Federico Iovino.

**Supervision:** Federico Iovino.

**Validation:** Mahebali Tabusi, Sigrun Thorsdottir, Maria Lysandrou, Ana Rita Narciso, Melania Minoia, Chinmaya Venugopal Srambickal, Jerker Widengren, Birgitta Henriques-Normark, Federico Iovino.

**Visualization:** Mahebali Tabusi, Sigrun Thorsdottir, Maria Lysandrou, Ana Rita Narciso, Melania Minoia, Chinmaya Venugopal Srambickal, Jerker Widengren, Federico Iovino.

**Writing – original draft:** Mahebali Tabusi, Birgitta Henriques-Normark, Federico Iovino.

**Writing – review & editing:** Mahebali Tabusi, Sigrun Thorsdottir, Maria Lysandrou, Ana Rita Narciso, Melania Minoia, Chinmaya Venugopal Srambickal, Jerker Widengren, Birgitta Henriques-Normark, Federico Iovino.

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
