## [Decision Letter · Decision Letter 0]

29 Jan 2021

Dear Dr. Iovino,

Thank you very much for submitting your manuscript "Neuronal death in pneumococcal meningitis is triggered by pneumolysin and RrgA interactions with β-actin" for consideration at PLOS Pathogens. As with all papers reviewed by the journal, your manuscript was reviewed by members of the editorial board and by independent reviewers. The reviewers appreciated the attention to an important topic. Based on the reviews, we are likely to accept this manuscript for publication, providing that you modify the manuscript according to the review recommendations.

While all reviewers feel that the manuscript is considerably improved, reviewer 2 maintains concerns regarding the strength of data supporting interactions of Ply and RrgA with beta-actin. These concerns should be addressed experimentally. Reviewer 1 has minor comments that should be addressed in the discussion as well.

Sincerely,

Carlos Javier Orihuela, PhD

Associate Editor

PLOS Pathogens

Michael Wessels

Section Editor

PLOS Pathogens

Kasturi Haldar

Editor-in-Chief

PLOS Pathogens

orcid.org/0000-0001-5065-158X

Michael Malim

Editor-in-Chief

PLOS Pathogens

orcid.org/0000-0002-7699-2064

While all reviewers feel that the manuscript is considerable improved, reviewer 2 maintains concerns regarding the strength of data supporting interactions of Ply and RrgA with beta-actin. These concerns should be addressed experimentally. Reviewer 1 has minor comments that should be addressed in the discussion as well.

Reviewer Comments (if any, and for reference):

Reviewer's Responses to Questions

**Part I - Summary**

Reviewer #1: This manuscript investigates the role of pneumococcal interactions with neuronal ß actin in the course of brain damage in meningitis. It is well known that there are many mechanisms by which pneumococci can kill cells, including neurons. This manuscript describes a new one. The authors present a very plausible scenario that a pneumococcal pilus brings the bacteria close to the neuronal membrane and this increases the toxicity of the cytotoxin Pln; these steps involve interactions with cell surface ß actin. The experiments follow a logical sequence and the data is solid, well controlled and strongly support the conclusions.

Minor Comments:

1) Abstract: Mechanisms of neuronal damage in meningitis have been extensively studied in many labs for decades. While this paper presents a new one, it is perhaps inaccurate to begin the abstract with “little is known about mechanisms that lead to neuronal death”

2) Introduction: Studies from this laboratory group have suggested that pilus is present on only ~20% of clinical pneumococcal strains. This should be made clear and any data indicating if pili are more abundant in meningeal strains should be cited. This would place the new mechanism in the context of the likelihood that in a given patient, the deployment of strategies against the mechanisms of damage shown in this study may not be commonly operative.

3) The studies with ß actin are particularly strong and innovative. The careful and detailed microscopy is highly informative and supports the conclusions.

4) Most of the studies shown are with a neuronal cell line. There are many different neuronal types; which one does this cell represent? In clinical meningitis, it is known that hippocampal neurons are found to be damaged more often than cortical neurons. Does the mechanism described in this paper suggest a reason for this variable susceptibility?

Reviewer #2: The authors showed that neuronal cell death in pneumococcal meningitis is caused by the interaction of the pneumococcal virulence factor Ply and the pilli factor RrgA with beta-actin. Furthermore, they also showed that pneumococci promoted RrgA- and Ply-dependent actin depolymerization, leading to Ca2+ influx and cell death. The results in this paper are too indirect to support their hypothesis, and mechanistic approach to elucidate the significance of Ply- and RrgA- interaction with actin should be done.

Reviewer #3: The authors have taken great care in addressing the concerns of the reviewers, including providing additional data.

**Part II – Major Issues: Key Experiments Required for Acceptance**

Reviewer #1: None

Reviewer #2: 1. The direct interaction of Ply D4 and actin should be required, and the author should demonstrate whether single amino acid substituted actin-binding deficient Ply-complemented pneumococci fails to increase the amount of β-actin exposed to the cell surface and dampens the RrgA adherence to neuron, Ca2+ influx and cell death without pore-forming activity. It is critical issue to demonstrate their hypothesis directly.

2. Similarly, the author should demonstrate whether single amino acid substituted actin-binding deficient RrgA-complemented pneumococci fails to Ca2+ influx and cell death. It is critical issue to support their hypothesis directly.

Reviewer #3: None.

**Part III – Minor Issues: Editorial and Data Presentation Modifications**

Reviewer #1: Listed in Part 1

Reviewer #2: 1. Quality of images are too poor.

2. Fig. S12 should be included in main text.

Reviewer #3: None

PLOS authors have the option to publish the peer review history of their article (what does this mean?). If published, this will include your full peer review and any attached files.

Reviewer #1: **Yes: **Elaine Tuomanen

Reviewer #2: No

Reviewer #3: No
---

## [Decision Letter · Decision Letter 1]

28 Feb 2021

Dear Dr. Iovino,

We are pleased to inform you that your manuscript 'Neuronal death in pneumococcal meningitis is triggered by pneumolysin and RrgA interactions with β-actin' has been provisionally accepted for publication in PLOS Pathogens.

Best regards,

Carlos Javier Orihuela, PhD

Associate Editor

PLOS Pathogens

Michael Wessels

Section Editor

PLOS Pathogens

Kasturi Haldar

Editor-in-Chief

PLOS Pathogens

orcid.org/0000-0001-5065-158X

Michael Malim

Editor-in-Chief

PLOS Pathogens

orcid.org/0000-0002-7699-2064

Reviewer Comments (if any, and for reference):

Reviewer's Responses to Questions

**Part I - Summary**

Reviewer #1: The authors have addressed most of the requirements put forward by the reviewers.

Reviewer #2: (No Response)

**Part II – Major Issues: Key Experiments Required for Acceptance**

Reviewer #1: Completed as listed

Reviewer #2: All of my concerns have been addressed.

**Part III – Minor Issues: Editorial and Data Presentation Modifications**

Reviewer #1: (No Response)

Reviewer #2: None

PLOS authors have the option to publish the peer review history of their article (what does this mean?). If published, this will include your full peer review and any attached files.

Reviewer #1: No

Reviewer #2: No

---

## [Editor Report · Acceptance letter]

11 Mar 2021

Dear Dr. Iovino,

We are delighted to inform you that your manuscript, "Neuronal death in pneumococcal meningitis is triggered by pneumolysin and RrgA interactions with β-actin," has been formally accepted for publication in PLOS Pathogens.

Best regards,

Kasturi Haldar

Editor-in-Chief

PLOS Pathogens

orcid.org/0000-0001-5065-158X

Michael Malim

Editor-in-Chief

PLOS Pathogens

orcid.org/0000-0002-7699-2064